# Improving Transformer World Models for Data-Efficient RL

Antoine Dedieu [* 1]   Joseph Ortiz [* 1]   Xinghua Lou [1]   Carter Wendelken [1]   J. Swaroop Guntupalli [1]
Wolfgang Lehrach [1]   Miguel Lázaro-Gredilla [1]   Kevin Murphy [1]

## Abstract

We present an approach to model-based RL that achieves a new state of the art performance on the challenging Craftax-classic benchmark, an open-world 2D survival game that requires agents to exhibit a wide range of general abilities—such as strong generalization, deep exploration, and long-term reasoning. With a series of careful design choices aimed at improving sample efficiency, our MBRL algorithm achieves a reward of 69.66% after only 1M environment steps, significantly outperforming DreamerV3, which achieves 53.2%, and, for the first time, exceeds human performance of 65.0%. Our method starts by constructing a SOTA model-free baseline, using a novel policy architecture that combines CNNs and RNNs. We then add three improvements to the standard MBRL setup: (a) "Dyna with warmup", which trains the policy on real and imaginary data, (b) "nearest neighbor tokenizer" on image patches, which improves the scheme to create the transformer world model (TWM) inputs, and (c) "block teacher forcing", which allows the TWM to reason jointly about the future tokens of the next timestep.

## 1. Introduction

Reinforcement learning (RL) (Sutton & Barto, 2018) provides a framework for training agents to act in environments so as to maximize their rewards. Online RL algorithms interleave taking actions in the environment—collecting observations and rewards—and updating the policy using the collected experience. Online RL algorithms often employ a model-free approach (MFRL), where the agent learns a direct mapping from observations to actions, but this can require a lot of data to be collected from the environment.

*Equal contribution [1]Google DeepMind. Correspondence to: Antoine Dedieu <adedieu@google.com>, Joseph Ortiz <joeortiz@google.com>.

*Proceedings of the 42nd International Conference on Machine Learning*, Vancouver, Canada. PMLR 267, 2025. Copyright 2025 by the author(s).

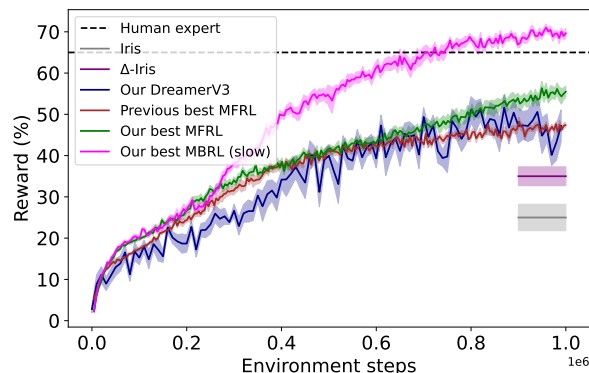

Figure 1: Reward on Craftax-classic. Our best MBRL and MFRL agents outperform all the previously published MFRL and MBRL results, and for the first time, surpass the reward achieved by a human expert. We display published methods which report the reward at 1M steps with horizontal line from 900k to 1M steps.

Model-based RL (MBRL) aims to reduce the amount of data needed to train the policy by also learning a world model (WM), and using this WM to plan "in imagination".

To evaluate sample-efficient RL algorithms, it is common to use the Atari-100k benchmark (Kaiser et al., 2019). However, although the benchmark encompasses a variety of skills (memory, planning, etc), each individual game typically only emphasizes one or two such skills. To promote the development of agents with broader capabilities, we focus on the Crafter domain (Hafner, 2021), a 2D version of Minecraft that challenges a single agent to master a diverse skill set. Specifically, we use the Craftax-classic environment (Matthews et al., 2024), a fast, near-replica of Crafter, implemented in JAX (Bradbury et al., 2018). Key features of Craftax-classic include: (a) procedurally generated stochastic environments (at each episode the agent encounters a new environment sampled from a common distribution); (b) partial observability, as the agent only sees a $63 \times 63$ pixel image representing a local view of the agent's environment, plus a visualization of its inventory (see Figure 2[left]); and (c) an achievement hierarchy that defines a sparse reward signal, requiring deep and broad exploration.

In this paper, we study improvements to MBRL methods, based on transformer world models (TWM), in the context

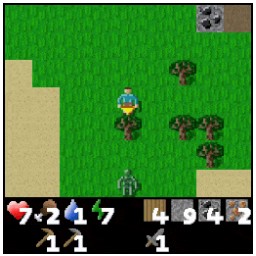 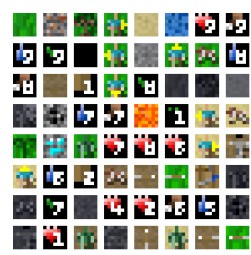

Figure 2: [Left] The Craftax-classic observation is a $63 \times 63$ pixel image, composed of $9 \times 9$ patches of $7 \times 7$ pixels. The observation shows the map around the agent and the agent's health and inventory. Here we have rendered the image at $144 \times 144$ pixels for visibility. [Right] 64 different patches.

of the Craftax-classic environment. We make contributions across the following three axes: (a) how the TWM is used (Section 3.4); (b) the tokenization scheme used to create TWM inputs (Section 3.5); (c) and how the TWM is trained (Section 3.6). Collectively, our improvements result in an agent that, with only 1M environment steps, achieves a Craftax-classic reward of 69.66% and a score of 31.77%, significantly improving over the previous state of the art (SOTA) reward of 53.20% (Hafner et al., 2023) and the previous SOTA score of 19.4% (Kauvar et al., 2023)[1].

Our first contribution relates to the way the world model is used: in contrast to recent MBRL methods like IRIS (Micheli et al., 2022) and DreamerV3 (Hafner et al., 2023), which train the policy solely on imagined trajectories (generated by the world model), we train our policy using both imagined rollouts from the world model and real experiences collected in the environment. This is similar to the original Dyna method (Sutton, 1990), although this technique has been abandoned in recent work. In this hybrid regime, we can view the WM as a form of generative data augmentation (Van Hasselt et al., 2019).

Our second contribution addresses the tokenizer which converts between images and tokens that the TWM ingests and outputs. Most prior work uses a vector quantized variational autoencoder (VQ-VAE, Van Den Oord et al. 2017), e.g. IRIS (Micheli et al., 2022), DART (Agarwal et al., 2024). These methods train a CNN to process images into a fea-

ture map, whose elements are then quantized into discrete tokens, using a codebook. The sequence of observation tokens across timesteps is used, along with the actions and rewards, to train the WM. We propose two improvements to the tokenizer. First, instead of jointly quantizing the image, we split the image into patches and independently tokenize each patch. Second, we replace the VQ-VAE with a simpler nearest-neighbor tokenizer (NNT) for patches. Unlike VQ-VAE, NNT ensures that the "meaning" of each code in the codebook is constant through training, which simplifies the task of learning a reliable WM.

Our third contribution addresses the way the world model is trained. TWMs are trained by maximizing the log likelihood of the sequence of tokens, which is typically generated autoregressively both over time and within a timeslice. We propose an alternative, which we call block teacher forcing (BTF), that allows TWM to reason jointly about the possible future states of all tokens within a timestep, before sampling them in parallel and independently given the history. With BTF, imagined rollouts for training the policy are both faster to sample and more accurate.

Our final contributions are some minor architectural changes to the MFRL baseline upon which our MBRL approach is based. These changes are still significant, resulting in a simple MFRL method that is much faster than Dreamer V3 and yet obtains a much better average reward and score.

Our improvements are complementary to each other, and can be combined into a "ladder of improvements"—similar to the "Rainbow" paper's (Hessel et al., 2018) series of improvements on top of model-free DQN agents.

## 2. Related Work

In this section, we discuss related work in MBRL — see e.g. Moerland et al. (2023); Murphy (2024); OpenDILab for more comprehensive reviews. We can broadly divide MBRL along two axes. The first axis is whether the world model (WM) is used for background planning (where it helps train the policy by generating imagined trajectories), or decision-time planning (where it is used for lookahead search at inference time). The second axis is whether the WM is a generative model of the observation space (potentially via a latent bottleneck) or whether is a latent-only model trained using a self-prediction loss (which is not sufficient to generate full observations).

Regarding the first axis, prominent examples of decision-time planning methods that leverage a WM include MuZero (Schrittwieser et al., 2020) and EfficientZero (Ye et al., 2021), which use Monte-Carlo tree search over a discrete action space, as well as TD-MPC2 (Hansen et al., 2024), which uses the cross-entropy method over a continuous action space. Although some studies have shown that decision-

---

[1]The score $S$ is given by the geometric mean of the success rate $s_i$ for each of the $N = 22$ achievements; this puts more weight on occasionally solving many achievements than on consistently solving a subset. More precisely, the score is given by $S = \exp\left(\frac{1}{N}\sum_{i=1}^{N}\ln(1 + s_i)\right) - 1$, where $s_i \in [0, 100]$ is the success percentage for achievement $i$ (i.e., fraction of episodes in which the achievement was obtained at least once). By contrast, the rewards are just the expected sum of rewards, or in percentage, the arithmetic mean $R = \frac{1}{N}\sum_{i=1}^{N} s_i$ (ignoring minor contributions to the reward based on the health of the agent). The score and reward are correlated, but are not the same. Unlike some prior work, we report both metrics to make comparisons easier.

time planning can sometimes be better than background planning (Alver & Precup, 2024), it is much slower, especially with large WMs such as transformers, since it requires rolling out future hypothetical trajectories at each decision-making step. Therefore in this paper, we focus on background planning (BP). Background planning originates from Dyna (Sutton, 1990), which focused on tabular Q-learning. Since then, many papers have combined the idea with deep RL methods: World Models (Ha & Schmidhuber, 2018b), Dreamer agents (Hafner et al., 2020a;b; 2023), SimPLe (Kaiser et al., 2019), IRIS (Micheli et al., 2022), Δ-IRIS (Micheli et al., 2024), Diamond (Alonso et al., 2024), DART (Agarwal et al., 2024), etc.

Regarding the second axis, many methods fit generative WMs of the observations (images) using a model with low-dimensional latent variables, either continuous (as in a VAE) or discrete (as in a VQ-VAE). This includes our method and most background planning methods above [2]. In contrast, other methods fit non-generative WMs, which are trained using self-prediction loss—see Ni et al. (2024) for a detailed discussion. Non-generative WMs are more lightweight and therefore well-suited to decision-time planning with its large number of WM calls at every decision-making step. However, generative WMs are generally preferred for background planning, since it is easy to combine real and imaginary data for policy learning, as we show below.

In terms of the architecture of the WM, many state-of-the-art models use transformers, e.g. IRIS (Micheli et al., 2022), Δ-IRIS (Micheli et al., 2024), DART (Agarwal et al., 2024). Notable exceptions are DreamerV2/3 (Hafner et al., 2020b; 2023), which use recurrent state space models, although improved transformer variants have been proposed (Robine et al., 2023; Zhang et al., 2024; Chen et al., 2022).

## 3. Methods

Here, we describe the components of our system, each of which improves performance, as we show in Section 4.

### 3.1. MFRL Baseline

Our starting point is the previous SOTA MFRL approach which was proposed as a baseline in Moon et al. (2024)[3]. This method achieves a reward of $46.91\%$ and a score of $15.60\%$ after 1M environment steps. This approach trains a stateless CNN policy without frame stacking using the PPO method (Schulman et al., 2017), and adds an entropy

penalty to ensure sufficient exploration. The CNN used is a modification of the Impala ResNet (Espeholt et al., 2018a).

### 3.2. MFRL Improvements

We improve on this MFRL baseline by both increasing the model size and adding a RNN (specifically a GRU) to give the policy memory. Interestingly, we find that naively increasing the model size harms performance, while combining a larger model with a carefully designed RNN helps (see Section 4.3). When varying the ratio of the RNN state dimension to the CNN encoder dimension, we observe that performance is higher when the hidden state is low-dimensional. Our intuition is that the memory is forced to focus on the relevant bits of the past that cannot be extracted from the current image.

We concatenate the GRU output to the image embedding, and then pass this to the actor and critic networks, rather than directly passing the GRU output. Algorithm 2, Appendix A.1, presents a pseudocode for our MFRL agent.

With these architectural changes, we increase the reward to $55.49\%$ and the score to $16.77\%$. This result is notable since our MFRL agent beats the considerably more complex (and much slower) DreamerV3 agent, which obtains a reward of $53.20\%$ and a score of $14.5$. It also beats other MBRL methods, such as IRIS (Micheli et al., 2022) (reward of $25.0\%$) and Δ-IRIS (Micheli et al., 2024) [4] (reward of $35.0\%$). In addition, our MFRL agent only takes 15 minutes to train for 1M environment steps on one A100 GPU.

### 3.3. MBRL baseline

We now describe our MBRL baseline, which combines our MFRL baseline above with a transformer world model (TWM)—as in IRIS (Micheli et al., 2022). Following IRIS, our MBRL baseline uses a VQ-VAE, which quantizes the $8 \times 8$ feature map $Z_t$ of a CNN to create a set of latent codes, $(q_t^1, \ldots, q_t^L) = \text{enc}(O_t)$, where $L = 64$, $q_t^i \in \{1, \ldots, K\}$ is a discrete code, and $K = 512$ is the size of the codebook. These codes are then passed to a TWM, which is trained using teacher forcing—see Equation (2) below. Our MBRL baseline achieves a reward of $31.93\%$, and improves over the reported results of IRIS, which reaches $25.0\%$.

Although these MBRL baselines leverage recent advances in generative world modeling, they are largely outperformed by our best MFRL agent. This motivates us to enhance our MBRL agent, which we explore in the following sections.

---

[2]A notable exception is Diamond (Alonso et al., 2024), which fits a diffusion world model directly in pixel space, rather than learning a latent WM.

[3]The authors' main method uses external knowledge about the achievement hierarchy of Crafter, so cannot be compared with other general methods. We use their baseline instead.

[4]This is consistent with results on Atari-100k, which show that well-tuned model-free methods, such as BBF (Schwarzer et al., 2023), can beat more sophisticated model-based methods.

## 3.4. MBRL using Dyna with warmup

As discussed in Section 1, we propose to train our MBRL agent on a mix of real trajectories (from the environment) and imaginary trajectories (from the TWM), similar to Dyna (Sutton, 1990). Algorithm 1 presents the pseudocode for our MBRL approach. Specifically, unlike many other recent MBRL methods (Ha & Schmidhuber, 2018a; Micheli et al., 2022; 2024; Hafner et al., 2020b; 2023) which train their policies exclusively using world model rollouts (Step 4), we include Step 2 which updates the policy with real trajectories. Note that, if we remove Steps 3 and 4 in Algorithm 1, the approach reduces to MFRL. The function rollout($O_1, \pi_\Phi, T, \mathcal{M}$) returns a trajectory of length $T$ generated by rolling out the policy $\pi_\Phi$ from the initial state $O_1$ in either the true environment $\mathcal{M}_{\text{env}}$ or the world model $\mathcal{M}_\Theta$. A trajectory contains collected observations, actions and rewards during the rollout $\tau = (O_{1:T+1}, a_{1:T}, r_{1:T})$. Algorithm 4 in Appendix A.3 details the rollout procedure. We discuss other design choices below.

**PPO.** Since PPO (Schulman et al., 2017) is an on-policy algorithm, trajectories should be used for policy updates immediately after they are collected or generated. For this reason, policy updates with real trajectories take place in Step 2 immediately after the data is collected. An alternative approach is to use an off-policy algorithm and mix real and imaginary data into the policy updates in Step 4, hence removing Step 2. We leave this direction as future work.

**Rollout horizon.** We set $T_{\text{WM}} \ll T_{\text{env}}$, to avoid the problem of compounding errors due to model imperfections (Lambert et al., 2022). However, we find it beneficial to use $T_{\text{WM}} \gg 1$, consistent with Holland et al. (2018); Van Hasselt et al. (2019), who observed that the Dyna approach with $T_{\text{WM}} = 1$ is no better than MFRL with experience replay.

**Multiple updates.** Following IRIS, we update TWM $N_{\text{WM}}^{\text{iters}}$ times and the policy on imagined trajectories $N_{\text{AC}}^{\text{iters}}$ times.

**Warmup.** When mixing imaginary trajectories with real ones, we need to ensure the WM is sufficiently accurate so that it does not harm policy learning. Consequently, we only begin training the policy on imaginary trajectories after the agent has interacted with the environment for $T_{\text{BP}}$ steps, which ensures it has seen enough data to learn a reliable WM. We call this technique "Dyna with warmup". In Section 4.3, we show that removing this warmup, and using $T_{\text{BP}} = 0$, drops the reward dramatically, from $67.42\%$ to $33.54\%$. We additionally show that removing the Dyna method (and only training the policy in imagination) drops the reward to $55.02\%$.

## 3.5. Patch nearest-neighbor tokenizer

Many MBRL methods based on TWMs use a VQ-VAE to map between images and tokens. In this section, we de-

---

**Algorithm 1** MBRL agent. See Appendix A.3 for details.

**Input:** number of environments $N_{\text{env}}$,
environment dynamics $\mathcal{M}_{\text{env}}$,
rollout horizon for environment $T_{\text{env}}$ and for TWM $T_{\text{WM}}$,
background planning starting step $T_{\text{BP}}$,
total number of environment steps $T_{\text{total}}$,
number of TWM updates $N_{\text{WM}}^{\text{iters}}$ and policy updates $N_{\text{AC}}^{\text{iters}}$

**Initialize:** observations $O_1^n \sim \mathcal{M}_{\text{env}}$ for $n = 1 : N_{\text{env}}$,
data buffer $\mathcal{D} = \emptyset$,
TWM model $\mathcal{M}$ and parameters $\Theta$,
AC model $\pi$ and parameters $\Phi$,
number of environment steps $t = 0$.

**repeat**
  *// 1. Collect data from environment*
  $\tau_{\text{env}}^n = \text{rollout}(O_1^n, \pi_\Phi, T_{\text{env}}, \mathcal{M}_{\text{env}}), \ n = 1 : N_{\text{env}}$
  $\mathcal{D} = \mathcal{D} \cup \tau_{\text{env}}^{1:N} \ ; \ O_1^{1:N} = \tau_{\text{env}}^{1:N}[-1] \ ; \ t{+}{=} N_{\text{env}}T_{\text{env}}$

  *// 2. Update policy on environment data*
  $\Phi = \text{PPO-update-policy}(\Phi, \tau_{\text{env}}^{1:N})$

  *// 3. Update world model*
  **for** it $= 1$ **to** $N_{\text{WM}}^{\text{iters}}$ **do**
    $\tau_{\text{replay}}^n = \text{sample-trajectory}(\mathcal{D}, T_{\text{WM}}), \ n = 1 : N_{\text{env}}$
    $\Theta = \text{update-world-model}(\Theta, \tau_{\text{replay}}^{1:N_{\text{env}}})$
  **end for**

  *// 4. Update policy on imagined data*
  **if** $t \geq T_{\text{BP}}$ **then**
    **for** it $= 1$ **to** $N_{\text{AC}}^{\text{iters}}$ **do**
      $\tilde{O}_1^n = \text{sample-obs}(\mathcal{D}), \ n = 1 : N_{\text{env}}$
      $\tau_{\text{WM}}^n = \text{rollout}(\tilde{O}_1^n, \pi_\Phi, T_{\text{WM}}, \mathcal{M}_\Theta), \ n = 1 : N_{\text{env}}$
      $\Phi = \text{PPO-update-policy}(\Phi, \tau_{\text{WM}}^{1:N_{\text{env}}})$
    **end for**
  **end if**
**until** $t \geq T_{\text{total}}$

---

scribe our alternative which leverages a property of Craftax-classic: each observation is composed of $9 \times 9$ patches of size $7 \times 7$ each (see Figure 2[left]). Hence we propose to (a) factorize the tokenizer by patches and (b) use a simpler nearest-neighbor style approach to tokenize the patches.

**Patch factorization.** Unlike prior methods which process the full image $O$ into tokens $(q^1, \ldots, q^L) = \text{enc}(O)$, we first divide $O$ into $L$ non-overlapping patches $(p^1, \ldots, p^L)$ which are independently encoded into $L$ tokens:

$$(q^1, \ldots, q^L) = (\text{enc}(p^1), \ldots, \text{enc}(p^L)) \ .$$

To convert the discrete tokens back to pixel space, we just decode each token independently into patches, and rearrange to form a full image:

$$(\hat{p}^1, \ldots, \hat{p}^L) = (\text{dec}(q^1), \ldots, \text{dec}(q^L)) \ .$$

Factorizing the VQ-VAE on the $L = 81$ patches of each observation boosts performance from $43.36\%$ to $58.92\%$.

**Nearest-neighbor tokenizer.** On top of patch factorization, we propose a simpler nearest-neighbor tokenizer (NNT) to replace the VQ-VAE. The encoding operation for each patch $p \in [0,1]^{h \times w \times 3}$ is similar to a nearest neighbor classifier w.r.t the codebook. The difference is that, if the nearest neighbor is too far away, we add a new code equal to $p$ to the codebook. More precisely, let us denote $\mathcal{C}_{\mathrm{NN}} = \{e_1, \ldots, e_K\}$ the current codebook, consisting of $K$ codes $e_i \in [0,1]^{h \times w \times 3}$, and $\tau$ a threshold on the Euclidean distance. The NNT encoder is defined as:

$$q = \mathrm{enc}(p) = \begin{cases} \underset{1 \le i \le K}{\mathrm{argmin}} \|p - e_i\|_2^2 & \text{if } \underset{1 \le i \le K}{\min} \|p - e_i\|_2^2 \le \tau \\ K+1 & \text{otherwise.} \end{cases}$$
(1)

The codebook can be thought of as a greedy approximation to the coreset of the patches seen so far (Mirzasoleiman et al., 2020). To decode patches, we simply return the code associated with the codebook index, i.e. $\mathrm{dec}(q^i) = e_{q^i}$.

A key benefit of NNT is that once codebook entries are added, they are never updated. A static yet growing codebook makes the target distribution for the TWM stationary, greatly simplifying online learning for the TWM. In contrast, the VQ-VAE codebook is continually updated, meaning the TWM must learn from a non-stationary distribution, which results in a worse WM. Indeed, we show in Section 4.1 that with patch factorization, and when $h = w = 7$—meaning that the patches are aligned with the observation—replacing the VQ-VAE with NNT boosts the agent's reward from $58.92\%$ to $64.96\%$. Figure 2[right] shows an example of the first 64 code patches extracted by our NNT.

The main disadvantages of our approach are that (a) patch tokenization can be sensitive to the patch size (see Figure 5[left]), and (b) NNT may create a large codebook if there is a lot of appearance variation within patches. In Craftax-classic, these problems are not very severe due to the grid structure of the game and limited sprite vocabulary (although continuous variations exist due to lighting and texture randomness).

### 3.6. Block teacher forcing

Transformer WMs are typically trained by teacher forcing which maximizes the log likelihood of the token sequence generated autoregressively over time and within a timeslice:

$$\mathcal{L}_{\mathrm{TF}} = \log \prod_{t=1}^{T} \prod_{i=1}^{L} \mathcal{L}_t^i, \quad \mathcal{L}_t^i = p(q_{t+1}^i | q_{1:t}^{1:L}, q_{t+1}^{1:i-1}, a_{1:t})$$
(2)

We propose a more effective alternative, which we call block teacher forcing (BTF). BTF modifies both the supervision and the attention of the TWM. Given the tokens from the

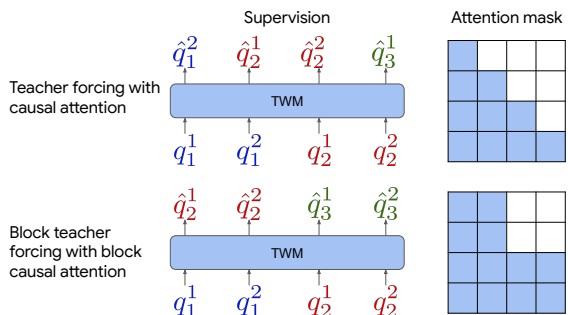

Figure 3: Approaches for TWM training with $L = 2, T = 2$. $q_t^\ell$ denotes token $\ell$ of timestep $t$. Tokens in the same timestep have the same color. We exclude action tokens for simplicity. [Top] Usual autoregressive model training with teacher forcing. [Bottom] Block teacher forcing predicts token $q_{t+1}^\ell$ from input token $q_t^\ell$ with block causal attention.

previous timesteps, BTF independently predicts all the latent tokens at the next timestep, removing the conditioning on previously generated tokens from the current step:

$$\mathcal{L}_{\mathrm{BTF}} = \log \prod_{t=1}^{T} \prod_{i=1}^{L} \tilde{\mathcal{L}}_t^i, \quad \tilde{\mathcal{L}}_t^i = p(q_{t+1}^i | q_{1:t}^{1:L}, a_{1:t})$$
(3)

Importantly BTF uses a block causal attention pattern (see Figure 3), in which tokens within the same timeslice are decoded in-parallel in a single forward pass. This attention structure allows the model to reason jointly about the possible future states of all tokens within a timestep, before sampling the tokens with independent readouts. This property mitigates autoregressive drift. As a result, BTF returns more accurate TWMs than fully AR approaches. Overall, adding BTF increases the reward from $64.96\%$ to $67.42\%$. In addition, we find that BTF is twice as fast, even though in theory, with key-value caching, BTF and AR both have complexity $\mathcal{O}(L^2 T)$ for generating all the $L$ tokens at one timestep, and $\mathcal{O}(L^2 T^2)$ for generating the entire rollout. Finally, BTF shares a similarity with Retentive Environment Models (REMs) (Cohen et al., 2024) in their joint prediction of next-frame tokens. However, while REMs employ a retentive network (Sun et al., 2023), BTF offers broader applicability across any transformer architecture.

## 4. Results

In this section, we report our experimental results on the Craftax-classic benchmark. Each experiment is run on 8 H100 GPUs. All methods are compared after interacting with the environment for $T_{\mathrm{total}} = 1M$ steps. All the methods collect trajectories of length $T_{\mathrm{env}} = 96$ in $N_{\mathrm{env}} = 48$ environment (in parallel). For MBRL methods, the imaginary rollouts are of length $T_{\mathrm{WM}} = 20$, and we start generating these (for policy training) after $T_{\mathrm{BP}} = 200k$ environment steps. We update the TWM $N_{\mathrm{WM}}^{\mathrm{iters}} = 500$ times and the policy $N_{\mathrm{AC}}^{\mathrm{iters}} = 150$ times. For all metrics, we report the

Table 1: Results on Craftax-classic after 1M environment interactions. * denotes results on Crafter, which may not exactly match Craftax-classic. — means unknown. †denotes the reported timings on a single A100 GPU. Our DreamerV3 results are based on the code from the author, but differ slightly from the reported number, perhaps due to hyperparameter discrepancies. IRIS and Δ-IRIS do not report standard errors for the score.

| Method | Parameters | Reward (%) | Score (%) | Time (min) |
|---|---|---|---|---|
| Human Expert | NA | $*65.0 \pm 10.5$ | $*50.5 \pm 6.8$ | NA |
| M1: Baseline | 60.0M | $31.93 \pm 2.22$ | $4.98 \pm 0.50$ | 560 |
| M2: M1 + Dyna | 60.0M | $43.36 \pm 1.84$ | $8.85 \pm 0.63$ | 563 |
| M3: M2 + patches | 56.6M | $58.92 \pm 1.03$ | $19.36 \pm 1.42$ | 746 |
| M4: M3 + NNT | 58.5M | $64.96 \pm 1.13$ | $25.55 \pm 0.86$ | 1328 |
| M5: M4 + BTF. Our best MBRL (fast) | 58.5M | $67.42 \pm 0.55$ | $27.91 \pm 0.63$ | 759 |
| M5: M4 + BTF. Our best MBRL (slow) | 58.5M | $\mathbf{69.66} \pm 1.20$ | $\mathbf{31.77} \pm 1.43$ | 2749 |
| Previous best MFRL (Moon et al., 2024) | 4.0M | $*46.91 \pm 2.41$ | $*15.60 \pm 1.66$ | — |
| Previous best MFRL (our implementation) | 4.0M | $47.40 \pm 0.58$ | $10.71 \pm 0.29$ | 26 |
| Our best MFRL | 55.6M | $55.49 \pm 1.33$ | $16.77 \pm 1.11$ | 15 |
| DreamerV3 (Hafner et al., 2023) | 201M | $*53.2 \pm 8.$ | $*14.5 \pm 1.6$ | — |
| Our DreamerV3 | 201M | $47.18 \pm 3.88$ | — | 2100 |
| IRIS (Micheli et al., 2022) | 48M | $*25.0 \pm 3.2$ | $*6.66$ | †8330 |
| Δ-IRIS (Micheli et al., 2024) | 25M | $*35.0 \pm 3.2$ | $*9.30$ | †833 |
| Curious Replay (Kauvar et al., 2023) | — | — | $*19.4 \pm 1.6$ | — |

mean and standard error over 10 seeds as $x(\pm y)$.

## 4.1. Climbing up the MBRL ladder

First, we report the normalized reward (the reward divided by the maximum reward of 22) for a series of agents that progressively climb our "MBRL ladder" of improvements in Section 3. Figure 4 shows the reward vs. the number of environment steps for the following methods, which we detail in Appendix A.2:

• **M1: Baseline**. Our baseline MBRL agent, described in Section 3.3, reaches a reward of 31.93%, and improves over IRIS, which gets 25.0%.

• **M2: M1 + Dyna**. Training the policy on both (real) environment and (imagined) TWM trajectories, as described in Section 3.4, increases the reward to 43.36%.

• **M3: M2 + patches**. Factorizing the VQ-VAE over the $L = 81$ observation patches, as presented in Section 3.5, increases the reward to 58.92%.

• **M4: M3 + NNT**. With patch factorization, replacing the VQ-VAE with NNT, as presented in Section 3.5, further boosts the reward to 64.96%.

• **M5: M4 + BTF. Our best MBRL (fast)**: Incorporating BTF, as described in Section 3.6, leads to our best agent. It achieves a reward of 67.42%, while BTF reduces the training time by a factor of two.

• **M5: M4 + BTF. Our best MBRL (slow)**: By increasing the number of TWM training steps to $N_{\text{WM}}^{\text{iters}} = 4\text{k}$, we obtain our best agent, which reaches a reward of 69.66%. However, due to substantial training times ($\sim$ 2 days), we do not include this agent in our ablation studies (Section 4.3) and comparative studies (Section 4.4).

As in IRIS (Micheli et al., 2022), methods M1-3 use a

codebook size of 512. For M4 and M5, which use NNT, we found it critical to use a larger codebook size of $K = 4096$ and a threshold of $\tau = 0.75$. Interestingly, when training in imagination begins (at step $T_{\text{BP}} = 200\text{k}$), there is a temporary drop in performance as the TWM rollouts do not initially match the true environment dynamics, resulting in a distribution shift for the policy.

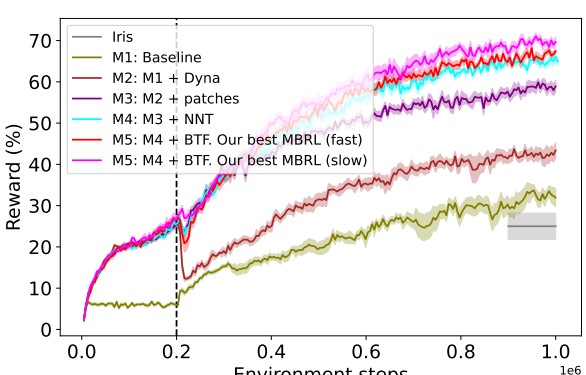

Figure 4: The ladder of improvements presented in Section 3 progressively transforms our baseline MBRL agent into a state-of-the-art method on Craftax-classic, reaching a reward of 69.66 (averaged over 10 seeds) after 1M environment steps. Training in imagination starts at step 200k, indicated by the dotted vertical line.

## 4.2. Comparison to existing methods

Figure 1 compares the performance of our best MBRL and MFRL agents against various previous methods. See also Figure 9 in Appendix B for a plot of the score, and Table 1 for a detailed numerical comparison of the final performance. First, we observe that our best MFRL agent outperforms

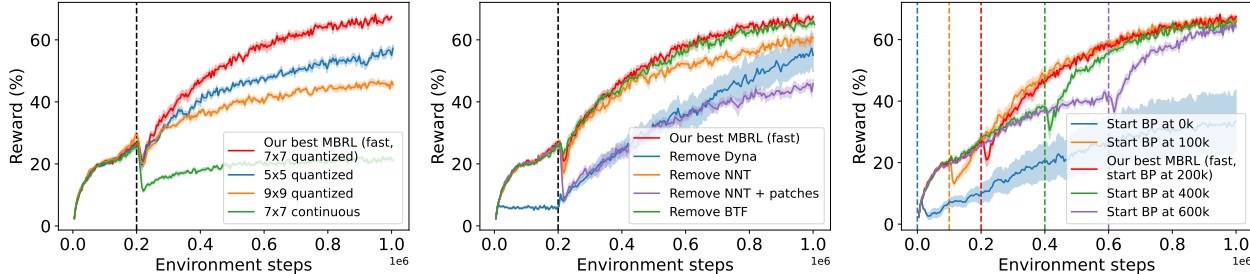

Figure 5: [Left] MBRL performance decreases when NNT uses patches of smaller or larger size than the ground truth, but it remains competitive. However, performance collapses if the patches are not quantized. [Middle] Removing any rung of the ladder of improvements leads to a drop in performance. [Right] Warming up the world model before using it to train the policy on imaginary rollouts is required for good performance. BP denotes background planning. For each method, training in imagination starts at the color-coded vertical line, and leads to an initial drop in performance.

almost all of the previously published MFRL and MBRL results, reaching a reward of $55.49\%$ and a score of $16.77\%$[5]. Second, our best MBRL agent achieves a new SOTA reward of $69.66\%$ and a score of $31.77\%$. This marks the first agent to surpass human-level reward, derived from 100 episodes played by 5 human expert players (Hafner, 2021). Note that although we achieve superhuman reward, our score is significantly below that of a human expert.

### 4.3. Ablation studies

We conduct ablation studies to assess the importance of several components of our proposed MBRL agent. Results are presented in Figure 5 and Table 2. All the TWMs are trained for $N_{\text{WM}}^{\text{iters}} = 500$ steps.

**Impact of patch size.** We investigate the sensitivity of our approach to the patch size used by NNT. While our best results are achieved when the tokenizer uses the oracle-provided ground truth patch size of $7 \times 7$, Figure 5[left] shows that performance remains competitive when using smaller ($5 \times 5$) or larger ($9 \times 9$) patches.

**The necessity of quantizing.** Figure 5[left] shows that, when the $7 \times 7$ patches are not quantized, but instead the TWM is trained to reconstruct the continuous $7 \times 7$ patches, MBRL performance collapses. This is consistent with findings in DreamerV2 (Hafner, 2021), which highlight that quantization is critical for learning an effective world model.

**Each rung matters.** To isolate the impact of each individual improvement, we remove each individual "rung" of our ladder from our best MBRL agent. As shown in Figure 5[middle], each removal leads to a performance drop. This underscores the importance of combining all our proposed enhancements to achieve SOTA performance.

[5]The only exception is Curious Replay (Kauvar et al., 2023), which builds on DreamerV3 with prioritized experience replay (PER) to train the WM. However, PER is only better on a few achievements; this improves the score but not the reward.

**When to start training in imagination?** Training the policy on imaginary TWM rollouts requires a reasonably accurate world model. This is why background planning (Step 4 in Algorithm 1) only begins after $T_{\text{BP}}$ environment steps. Figure 5[right] explores the effect of varying $T_{\text{BP}}$. Initiating imagination training too early ($T_{\text{BP}} = 0$) leads to performance collapse due to the inaccurate TWM dynamics.

**MFRL ablation.** The final 3 rows in Table 2 show that either removing the RNN or using a smaller model as in Moon et al. (2024) leads to a drop in performance.

Table 2: Ablations results.

| Method | Reward (%) | Score (%) |
|---|---|---|
| Our best MBRL (fast) | $\mathbf{67.42} \pm 0.55$ | $\mathbf{27.91} \pm 0.63$ |
| $5 \times 5$ quantized | $57.28 \pm 1.14$ | $18.26 \pm 1.18$ |
| $9 \times 9$ quantized | $45.55 \pm 0.88$ | $10.12 \pm 0.40$ |
| $7 \times 7$ continuous | $21.20 \pm 0.55$ | $2.43 \pm 0.09$ |
| Remove Dyna | $55.02 \pm 5.34$ | $18.79 \pm 2.14$ |
| Remove NNT | $60.66 \pm 1.38$ | $21.79 \pm 1.33$ |
| Remove NNT & patches | $45.86 \pm 1.42$ | $10.36 \pm 0.69$ |
| Remove BTF | $64.96 \pm 1.13$ | $25.55 \pm 0.86$ |
| Use $T_{\text{BP}} = 0$ | $33.54 \pm 10.09$ | $12.86 \pm 4.05$ |
| Best MFRL | $55.49 \pm 1.33$ | $16.77 \pm 1.11$ |
| Remove RNN | $41.82 \pm 0.97$ | $8.33 \pm 0.44$ |
| Smaller model | $51.35 \pm 0.80$ | $12.93 \pm 0.56$ |

**Annealing the number of policy updates:** We linearly increase the number of policy updates on imaginary rollouts in Step 4 of Algorithm 1 from $N_{\text{AC}}^{\text{iters}} = 0$ (when $T_{\text{total}} = 0$) to $N_{\text{AC}}^{\text{iters}} = 300$ (when $T_{\text{total}} = 1M$). This annealing technique achieves a reward of $65.71\%(\pm 1.11)$, while removing the drop in performance observed when we start training in imagination. See Figure 10 Appendix C.

### 4.4. Comparing TWM rollouts

In this section, we compare the TWM rollouts learned by three world models in our ladder, namely M1, M3 and our

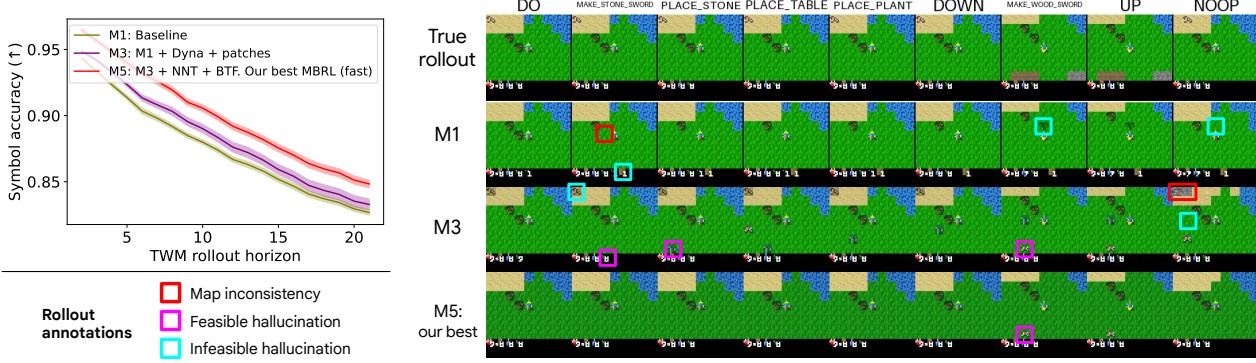

Figure 6: Rollout comparison for world models M1, M3 and M5 (fast). [Left] Symbol accuracies decrease with the TWM rollout step. The stationary NNT codebook used by M5 makes it easier to learn a reliable TWM. [Right] Best viewed zoomed in. **Map.** All three models accurately capture the agent's motion. All models can struggle to use the history to generate a consistent map when revisiting locations, however only M1 makes simple map errors in successive timesteps. **Feasible hallucinations.** M3 and M5 generate realistic hallucinations that respect the game dynamics, such as spawning mobs and losing health. **Infeasible hallucinations.** M1 often does not respect game dynamics; M1 incorrectly adds wood inventory, and incorrectly places a plant at the wrong timestep without the required sapling inventory. M3 exhibits some infeasible hallucinations in which the monster suddenly disappears or the spawned cow has an incorrect appearance. M5 rarely exhibits infeasible hallucinations. Figure 12 in Appendix D.4 shows more rollouts with similar behavior.

best model M5 (fast). To do so, we first create an evaluation dataset of $N_{\text{eval}} = 160$ trajectories, each of length $T_{\text{eval}} = T_{\text{WM}} = 20$, collected during the training of our best MFRL agent: $\mathcal{D}_{\text{eval}} = \left\{ O_{1:T_{\text{eval}}+1}^{1:N_{\text{eval}}}, a_{1:T_{\text{eval}}}^{1:N_{\text{eval}}}, r_{1:T_{\text{eval}}}^{1:N_{\text{eval}}} \right\}$. We evaluate the quality of imagined trajectories generated by each TWM. Given a TWM checkpoint at 1M steps and the $n$th trajectory in $\mathcal{D}_{\text{eval}}$, we execute the sequence of actions $a_{1:T_{\text{eval}}}^n$, starting from $O_1^n$, to obtain a rollout trajectory $\hat{O}_{1:T_{\text{eval}}+1}^{\text{TWM},\, n}$.

**Quantitative evaluations.** For evaluation, we leverage an appealing property of Craftax-classic: each observation $O_t$ comes with an array of ground truth symbols $S_t = (S_t^{1:R})$, with $R = 145$. Given 100k pairs $(O_t, S_t)$, we train a CNN $f_\mu$, to predict the symbols from the observation; $f_\mu$ achieves a 99% validation accuracy. Next, we use $f_\mu$ to predict the symbols from the generated rollouts. Figure 6[left] displays the average symbol accuracy at each timestep $t$:

$$\mathcal{A}_t = \frac{1}{N_{\text{eval}} R} \sum_{n=1}^{N_{\text{eval}}} \sum_{r=1}^{R} \mathbf{1}(f_\mu^r(\hat{O}_t^{\text{TWM},\, n}), S_t^{r,n}), \ \forall t,$$

where $\mathbf{1}(x, y) = 1$ iff. $x = y$ (and 0 o.w.), $S_t^{r,n}$ denotes the ground truth $r$th symbol in the array $S_t^n$ associated with $O_t^n$, and $f_\mu^r(\hat{O}_t^{\text{TWM},\, n})$ its prediction for the rollout observation. As expected, symbol accuracies decrease with $t$ as mistakes compound over the rollouts. Our best method, which uses NNT, achieves the highest accuracies for all timesteps, as it best captures the game dynamics. This highlights that a stationary codebook makes TWM learning simpler.

We include two additional quantitative evaluations in Appendix D, showing that M5 achieves the lowest tokenizer reconstruction errors and rollout reconstruction errors.

**Qualitative evaluations.** Due to environment stochasticity, TWM rollouts can differ from the environment rollout but

still be useful for learning in imagination—as long as they respect the game dynamics. Visual inspection of rollouts in Figure 6[right] reveals (a) map inconsistencies, (b) feasible hallucinations that respect the game dynamics and (c) infeasible hallucinations. M1 can make simple mistakes in both the map and the game dynamics. M3 and M5 both generate feasible hallucinations of mobs, however M3 more often hallucinates infeasible rollouts.

### 4.5. Craftax Full

Table 3: Results on Craftax after 1M environment interactions. The previous SOTA scores are unknown.

| Method | Reward (%) | Score (%) |
|---|---|---|
| Prev. SOTA MFRL | 2.3 (symbolic) | — |
| Our best MFRL | $4.63 \pm 0.20$ | $1.22 \pm 0.07$ |
| Prev. SOTA MBRL | 6.59 | — |
| Our best MBRL (slow) | $\mathbf{7.20 \pm 0.09}$ | $\mathbf{2.31 \pm 0.04}$ |

Table 3 compares the performance of various agents on the full version of Craftax (Matthews et al., 2024), a significantly harder extension of Craftax-classic, with more levels and achievements. While the previous SOTA agent reached 2.3% reward (on symbolic inputs), our MFRL agent reaches 4.63% reward. Similarly, while the recent SOTA MBRL (Cohen et al., 2025) reaches 6.59% reward our MBRL agent reaches a new SOTA reward of 7.20%. See Appendix E for implementation details.

### 4.6. Additional experiments on MinAtar

To further validate the robustness of our approach, we conduct additional experiments on MinAtar (Young & Tian, 2019), another grid world environment. MinAtar implements four simplified Atari 2600 games. Each game has

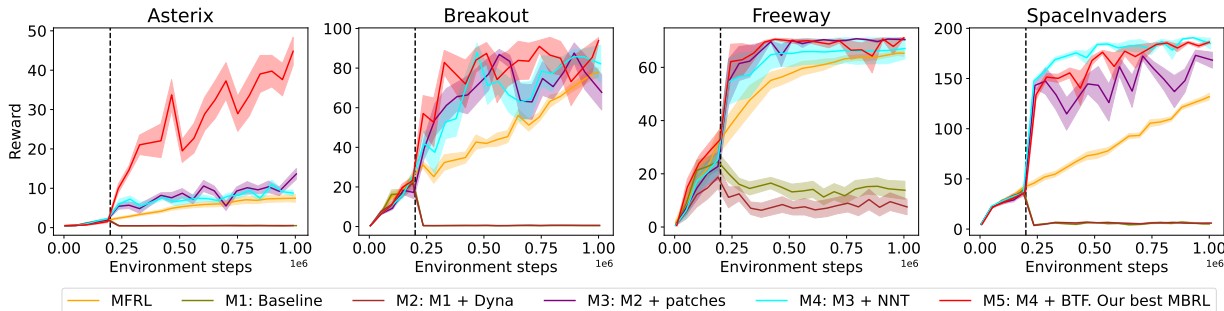

Figure 7: Our best MBRL agent leads to significant gains over our tuned MFRL agent on each MinAtar game.

symbolic binary observations of size $10 \times 10 \times K$ ($K$ is the number of objects of the game) and binary rewards.

We first tune our model-free RL agent on the MinAtar games, keeping the same architecture as described in our paper, with minor adjustments to the PPO hyperparameters, detailed in Appendix F. Second, we develop our model-based RL agent as in Craftax-classic, by integrating our three proposed improvements. We retain the majority of the MBRL hyperparameters from Craftax-classic, with minor modifications, which we detail in Appendix F.

Figure 7 displays the evaluation performance of our proposed methods M1-5 (defined as in Section 4.1) on each game after 1 million environment steps, averaged over 10 seeds. Every 50k training steps, we evaluate each agent on 32 environments and 2k steps per environments. Figure 8 summarizes these results by first (a) normalizing each game such that the MFRL agent achieves a reward of 1.0, before (b) averaging the performance of all agents across the games. Notably, our MBRL agents' performance increase as we climb the ladder on MinAtar, highlighting the generality of our three proposed improvements. Furthermore, our best MBRL agent significantly outperforms our best MFRL agent, achieving an average normalized reward of 2.41 across the four MinAtar games.

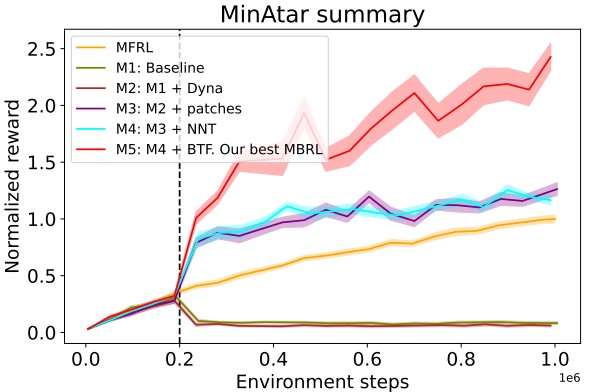

Figure 8: Averaged normalized reward on MinAtar.

Finally, Table 4 compares the performance of our best MBRL and MFRL agents at 1M steps, further emphasizing

the significant performance improvements achieved by our proposed MBRL agent.

Table 4: Best MFRL and best MBRL rewards after 1M steps on each MinAtar game.

| Game | MFRL | MBRL |
|---|---|---|
| Asterix | $7.47 \pm 1.02$ | $\mathbf{44.81} \pm 3.54$ |
| Breakout | $77.8 \pm 2.28$ | $\mathbf{93.92} \pm 1.44$ |
| Freeway | $65.3 \pm 1.16$ | $\mathbf{71.12} \pm 0.13$ |
| SpaceInvaders | $131.9 \pm 3.32$ | $\mathbf{186.16} \pm 1.25$ |

## 5. Conclusion and future work

In this paper, we present three improvements to vision-based MBRL agents which use transformer world models for background planning: Dyna with warmup, patch nearest-neighbor tokenization and block teacher forcing. We also present improvements to the MFRL baseline, which may be of independent interest. Collectively, these improvements result in a MBRL agent that achieves a significantly higher reward and score than previous SOTA agents on the challenging Craftax-classic benchmark, and surpasses expert human reward for the first time. Our improvements also transfer to MinAtar environments. In the future, we plan to examine how well our techniques generalize beyond grid-world environments. However, we believe our current results will already be of interest to the community.

We see several paths to build upon our method. Prioritized experience replay is a promising approach to accelerate TWM training, and an off-policy RL algorithm could improve policy updates by mixing imagined and real data. In the longer term, we would like to generalize our tokenizer to extract patches and tokens from large pre-trained models, such as SAM (Ravi et al., 2024) and Dino-V2 (Oquab et al., 2024). This inherits the stable codebook of our approach, but reduces sensitivity to patch size and "superficial" appearance variations. To explore this direction, and other non-reconstructive world models which cannot generate future pixels, we plan to modify the policy to directly accept latent tokens generated by the TWM.

## Acknowledgments

We thank Pablo Samuel Castro for useful discussions during the preparation of this manuscript.

## Impact Statement

This paper proposes new techniques for creating AI agents that can rapidly learn to play well in open-world environments. As our best agent surpasses human-level performance in a challenging benchmark game with limited interaction, this work paves the way for developing more sample-efficient and general-purpose AI agents capable of tackling real-world problems that demand strong generalization, exploration, and long-term reasoning.

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

# A. Algorithmic details

## A.1. Our Model-free RL agent

We first detail our new state-of-the-art MFRL agent. As mentioned in the main text, it relies on an actor-critic policy network trained with PPO.

### A.1.1. MFRL ARCHITECTURE

We summarize our MFRL agent in Algorithm 2 and further detail it below.

---

**Algorithm 2** MFRL agent

---

**Input:** Image $O_t$, last hidden state $h_{t-1}$, parameters $\Phi$.
**Output:** action $a_t$, value $v_t$, new hidden state $h_t$.
$z_t = \text{ImpalaCNN}_\Phi(O_t)$
$h_t, y_t = \text{RNN}_\Phi([h_{t-1}, z_t])$
$a_t \sim \pi_\Phi([y_t, z_t])$
$v_t = V_\Phi([y_t, z_t])$

---

**Imapala CNN architecture:** Each Craftax-classic image $O_t$ of size $63 \times 63 \times 3$ goes through an Impala CNN (Espeholt et al., 2018b). The CNN consists of three stacks with channel sizes of $(64, 64, 128)$. Each stack is composed of (a) batch normalization (Ioffe & Szegedy, 2015), (b) a convolutional layer with kernel size $3 \times 3$ and stride of 1, (c) a max pooling layer with kernel size $3 \times 3$ and stride of 2, and (d) two ResNet blocks (He et al., 2016). Each ResNet block is composed of (a) a ReLU activation followed by a batch normalization, (b) a convolutional layer with kernel size $3 \times 3$ and stride of 1. The CNN last layer output, of size $8 \times 8 \times 128$ passes through a ReLU activation, then gets flattened into an embedding vector of size 8192, which we call $z_t$.

**RNN architecture:** The CNN output $z_t$ (a) goes through a layer norm operator, (b) then gets linearly mapped to a 256-dimensional vector, (c) then passes through a ReLU activation, resulting in the new input for the RNN. The RNN then updates its hidden state, and outputs a 256-dimensional vector $y_t$, which goes through another ReLU activation.

**Actor and critic architecture:** Finally, the CNN output $z_t$ and the RNN output $y_t$ are concatenated, resulting in the 8448-dimensional embedding input shared by the actor and the critic networks. For the actor network, this shared input goes through (a) a layer normalization (Lei Ba et al., 2016), (b) a fully-connected network whose 2048-dimensional output goes through a ReLU, (c) two dense residual blocks whose 2048-dimensional output goes through a ReLU, (d) a last layer normalization and (e) a final fully-connected network which predicts the action logits.

Similarly, for the critic network, the shared input goes through (a) a layer normalization, (b) a fully-connected network whose 2048-dimensional output goes through a ReLU, (c) two dense residual blocks whose 2048-dimensional output goes through a ReLU, (d) a last layer normalization and (e) a final layer which predicts the value (which is a float).

### A.1.2. PPO TRAINING

We train our MFRL agent with the PPO algorithm (Schulman et al., 2017). PPO is a policy gradient algorithm, which we briefly summarize below.

**Training objective:** We assume we are given a trajectory, $\tau = (O_{1:T+1}, a_{1:T}, r_{1:T}, \text{done}_{1:T}, h_{0:T})$ collected in the environment, where $\text{done}_t$ is a binary variable indicating whether the current state is a terminal state, and $h_t$ is the RNN hidden state collected while executing the policy. Algorithm 4 discusses how we collect such a trajectory.

Given the fixed current actor-critic parameters $\Phi_{\text{old}}$, PPO first runs the actor-critic network on $\tau$, starting from the hidden state $h_0$ and returns two sequences of values $v_{1:T+1} = V_{\Phi_{\text{old}}}(O_{1:T+1})$ and probabilities $\pi_{\Phi_{\text{old}}}(a_t|O_t)$[6]. It then defines the generalized advantage estimation (GAE) as in Schulman et al. (2015):

$$A_t = \delta_t + (1 - \text{done}_t)\gamma\lambda A_{t+1} = \delta_t + (1 - \text{done}_t)\left(\gamma\lambda\delta_{t+1} + \ldots + (\gamma\lambda)^{T-t}\delta_T\right). \quad \forall t \leq T$$

---

[6]We drop the ImpalaCNN and the RNN for simplicity.

where
$$\delta_t = r_t + (1 - \text{done}_t)\gamma v_{t+1} - v_t.$$

PPO also defines the TD targets $q_t = A_t + v_t$.

PPO optimizes the parameters $\Phi$, to minimize the objective value:

$$\mathcal{L}_{\text{PPO}}(\Phi) = \frac{1}{T}\sum_{t=1}^{T}\left\{-\min\left(r_t(\Phi)A_t, \text{clip}(r_t(\Phi))A_t\right) + \lambda_{\text{TD}}(V_\Phi(O_t) - q_t)^2 - \lambda_{\text{ent}}\mathcal{H}(\pi_\Phi(.|O_t))\right\}, \tag{4}$$

where $\text{clip}(u)$ ensures that $u$ lies in the interval $[1 - \epsilon, 1 + \epsilon]$, $r_t(\Phi)$ is the probability ratio $r_t(\Phi) = \frac{\pi_\Phi(a_t|O_t)}{\pi_{\Phi\text{old}}(a_t|O_t)}$ and $\mathcal{H}$ is the entropy operator.

**Algorithm:** Algorithm 3 details the PPO-update-policy, which is called in Steps 1 and 4 in our main Algorithm 1 to update the PPO parameters on a batch of trajectories. PPO allows multiple epochs of minibatch updates on the same batch and introduces two hyperparameters: a number of minibatches $N^{\text{mb}}$ (which divides the number of environments $N_{\text{env}}$), and a number of epochs $N^{\text{epoch}}$.

---

**Algorithm 3** PPO-update-policy

---

**Input:** Actor-critic model $(\pi, V)$ and parameters $\Phi$
Trajectories $\tau^{1:N_{\text{env}}} = (O_{1:T+1}^{1:N_{\text{env}}}, a_{1:T}^{1:N_{\text{env}}}, r_{1:T}^{1:N_{\text{env}}}, \text{done}_{1:T}^{1:N_{\text{env}}}, h_{0:T}^{1:N_{\text{env}}})$
Number of epochs $N^{\text{epoch}}$ and of minibatches $N^{\text{mb}}$
PPO objective value parameters $\gamma, \lambda, \epsilon$
Learning rate lr and max-gradient-norm
Moving average mean $\mu_{\text{target}}$, standard deviation $\sigma_{\text{target}}$ and discount factor $\alpha$

**Output:** Updated actor-critic parameters $\Phi$

**Initialize:** Define $\Phi_{\text{old}} = \Phi$
Compute the values $v_{1:T+1}^{1:N_{\text{env}}} = V_{\Phi_{\text{old}}}(O_{1:T+1}^{1:N_{\text{env}}})$
Compute PPO GAEs and targets $A_{1:T}^{1:N_{\text{env}}}, q_{1:T}^{1:N_{\text{env}}} = \text{GAE}(r_{1:T}^{1:N_{\text{env}}}, v_{1:T+1}^{1:N_{\text{env}}}, \gamma, \lambda)$
Standardize PPO GAEs $A_{1:T}^{1:N_{\text{env}}} = \frac{A_{1:T}^{1:N_{\text{env}}} - \text{mean}(A_{1:T}^{1:N_{\text{env}}})}{\text{std}(A_{1:T}^{1:N_{\text{env}}})}$

**for** ep $= 1$ **to** $N^{\text{epoch}}$ **do**
  **for** $k = 1$ **to** $N^{\text{mb}}$ **do**
    $N^{\text{start}} = (k - 1)\left(N_{\text{env}}/N^{\text{mb}}\right) + 1, \quad N^{\text{end}} = k\left(N_{\text{env}}/N^{\text{mb}}\right) + 1$
    *// Standardize PPO target*
    Update $\mu_{\text{target}} = \alpha\mu_{\text{target}} + (1 - \alpha)\text{mean}(q_{1:T}^{N^{\text{start}}:N^{\text{end}}})$
    Update $\sigma_{\text{target}} = \alpha\sigma_{\text{target}} + (1 - \alpha)\text{std}(q_{1:T}^{N^{\text{start}}:N^{\text{end}}})$
    Standardize $q_{1:T}^{N^{\text{start}}:N^{\text{end}}} = (q_{1:T}^{N^{\text{start}}:N^{\text{end}}} - \mu_{\text{target}})/\sigma_{\text{target}}$

    *// Run the actor-critic network*
    Define $\tilde{h}_0^{N^{\text{start}}:N^{\text{end}}} = h_0^{N^{\text{start}}:N^{\text{end}}}$
    **for** $t = 1$ **to** $T + 1$ **do**
      $z_t^n = \text{ImpalaCNN}_\Phi(O_t^n) \;\; ; \;\; \tilde{h}_t^n = \text{RNN}_\Phi([\tilde{h}_{t-1}^n, z_t^n])$       **for** $n = N^{\text{start}} : N^{\text{end}}$
      Compute $V_\Phi^n([y_t^n, z_t^n])$ and $\pi_\Phi^n([y_t^n, z_t^n])$       **for** $n = N^{\text{start}} : N^{\text{end}}$
    **end for**

    *// Take a gradient step*
    Compute $\mathcal{L}_{\text{PPO}}^n(\Phi)$ using Equation (4)       **for** $n = N^{\text{start}} : N^{\text{end}}$
    Define the minibatch loss $\mathcal{L}_{\text{PPO}}(\Phi) = \frac{1}{N^{\text{mb}}}\sum_{n=N^{\text{start}}}^{N^{\text{end}}}\mathcal{L}_{\text{PPO}}^n(\Phi)$
    Update $\Phi = \text{Adam}\left(\Phi, \text{ clip-gradient}(\nabla_\Phi\mathcal{L}_{\text{PPO}}(\Phi), \text{max-norm}), \text{ lr}\right)$
  **end for**
**end for**

---

We make a few comments below:

• We use gradient clipping on each minibatch to control the maximum gradient norm, and update the actor-critic parameters using Adam (Kingma, 2014) with learning rate of 0.00045.

• During each epoch and minibatch update, we initialize the hidden state $\tilde{h}_0$ from its value $h_0$ stored while collecting the trajectory $\tau$.

• In Algorithm 3, we introduce two changes to the standard PPO objective, described in Equation (4). First, we standardize the GAEs (ensure they are zero mean and unit variance) across the batches. Second, similar to Moon et al. (2024), we maintain a moving average with discount factor $\alpha$ for the mean and standard deviation of the target $q_t$ and we update the value network to predict the standardized targets.

**Implementation:**   Note that for implementing PPO, we start from the code available in the `purejaxrl` library (Lu et al., 2022) at `https://github.com/luchris429/purejaxrl/blob/main/purejaxrl/ppo.py`.

A.1.3. HYPERPARAMETERS

Table 5 displays the PPO hyperparameters used for training our SOTA MFRL agent.

Table 5: MFRL hyperpameters

| Module | Hyperparameter | Value |
|---|---|---|
| Environment | Number of environments $N_{\text{env}}$ | 48 |
| | Rollout horizon in environment $T_{\text{env}}$ | 96 |
| Sizes | Image size | $63 \times 63 \times 3$ |
| | CNN output size | $8 \times 8 \times 128$ |
| | RNN hidden layer size | 256 |
| | AC input size | 8448 |
| | AC layer size | 2048 |
| PPO | $\gamma$ | 0.925 |
| | $\lambda$ | 0.625 |
| | $\epsilon$ clipping | 0.2 |
| | TD-loss coefficient $\lambda_{\text{TD}}$ | 1.0 |
| | Entropy loss coefficient $\lambda_{\text{ent}}$ | 0.01 |
| | PPO target discount factor $\alpha$ | 0.95 |
| Learning | Optimizer | Adam (Kingma, 2014) |
| | Learning rate | 0.00045 |
| | Max. gradient norm | 0.5 |
| | Learning rate annealing (MFRL) | True (linear schedule) |
| | Number of minibatches (MFRL) | 8 |
| | Number of epochs (MFRL) | 4 |

**MBRL experiments.** We make two additional changes to PPO in the MBRL setting, and keep all the other hyperparameters fixed. First, we do not use learning rate annealing for MBRL, while MFRL uses learning rate annealing (with a linear schedule). Second, as we discuss in Section A.3.3, the differences between the PPO updates on real and imaginary trajectories lead to varying the number of minibatches and epochs.

**Craftax experiments.**   We also keep all but two of our PPO hyperparameters fixed for Craftax (full), which we discuss in Appendix E.

## A.2. Model-based modules

In this section, we detail the two key modules for model-based RL: the tokenizer and the transformer world model.

### A.2.1. TOKENIZER

**Training objective:** Given a Craftax-classic image $O_t$ and a codebook $\mathcal{C} = \{e_1, \ldots, e_K\}$, an encoder $\mathcal{E}$ returns a feature map $Z_t = (Z_t^1, \ldots, Z_L^t)$. Each feature $Z_t^\ell$ gets quantized, resulting into $L$ tokens $Q_t = (q_t^1, \ldots, q_t^L)$—which serves as input to the TWM—then projected back to $\hat{Z}_t = (e_{q_t^1}, \ldots e_{q_t^L})$. Finally, a decoder $\mathcal{D}$ decodes $\hat{Z}_t$ back to the image space: $\hat{O}_t = \mathcal{D}(\hat{Z}_t)$. Following Van Den Oord et al. (2017); Micheli et al. (2022), we define the VQ-VAE loss as:

$$\mathcal{L}_{\text{VQ-VAE}}(\mathcal{E}, \mathcal{D}, \mathcal{C}) = \lambda_1 \|O_t - \hat{O}_t\|_1 + \lambda_2 \|O_t - \hat{O}_t\|_2^2 + \lambda_3 \|\text{sg}(Z_t) - \hat{Z}_t\|_2^2 + \lambda_4 \|Z_t - \text{sg}(\hat{Z}_t)\|_2^2, . \tag{5}$$

where sg is the stop-gradient operator. The first two terms are the reconstruction loss, the third term is the codebook loss and the last term is a commitment loss.

We now discuss the different VQ and VQ-VAE architectures used by the models M1-5 in the ladder described in Section 4.1.

**Default VQ-VAE:** Our baseline model M1, and our next model M2 build on IRIS VQ-VAE (Micheli et al., 2022) and follow the authors' code: `https://github.com/eloialonso/iris/blob/main/src/models/tokenizer/nets.py`. The encoder uses a convolutional layer (with kernel size $3 \times 3$ and stride 1), then five residual blocks with two convolutional layers each (with kernel size $3 \times 3$, stride 1 and ReLU activation). The channel sizes of the residual blocks are $(64, 64, 128, 128, 256)$. A downsampling is applied on the first, third and fourth blocks. Finally, a last convolutional layer with 128 channels returns an output of size $8 \times 8 \times 128$. The decoder follows the reverse architecture. Each of the $L = 64$ latent embeddings gets quantized individually, using a codebook of size $K = 512$, to minimize Equation (5). We use codebook normalization, meaning that each code in the codebook $\mathcal{C}$ has unit L2 norm. Similarly, each latent embedding $Z_t^\ell l$ gets normalized before being quantized. As in IRIS, we use $\lambda_1 = 1, \lambda_2 = 0, \lambda_3 = 1, \lambda_4 = 0.25$. We train with Adam and a learning rate of $0.001$.

**VQ-VAE(patches):** For the next model M3, the encoder is a two-layers MLP that maps each flattened $7 \times 7 \times 3$ patch to a 128-dimensional vector, using a ReLU activation. Similarly, the decoder learns a linear mapping from the embedding vector back to the flattened patches. Each embedding gets quantized individually, using a codebook of size $K = 512$, and codebook normalization, to minimize Equation (5). Following Micheli et al. (2024), we use $\lambda_1 = 0.1, \lambda_2 = 1, \lambda_3 = 1, \lambda_4 = 0.02$.

**Nearest neighbor tokenizer:** NNT does not use Equation (5) and directly adds image patches to a codebook of size $K = 4096$, using a Euclidean threshold $\tau = 0.75$.

### A.2.2. TRANSFORMER WORLD MODEL

**Training objective:** We train the TWM on real trajectories (from the environment) of $T_{\text{WM}} = 20$ timesteps sampled from the replay buffer (see Algorithm 1). We set $T_{\text{WM}} = 20$ as it is the largest value that will fit into memory on 8 H100 GPUs.

Given a trajectory $\tau = (O_{1:T+1}, a_{1:T}, r_{1:T}, \text{done}_{1:T})$, the input to the transformer is the sequence of tokens:

$$(q_1^1, \ldots, q_1^L, a_1, \ldots q_T^1, \ldots, q_T^L, a_T),$$

where $\text{enc}(O_t) = (q_t^1, \ldots, q_t^L)$ and $q_t^i \in \{1, \ldots, K\}$ where $K$ is the codebook size. These tokens are then embedded using an observation embedding table and an action embedding table. After several self-attention layers (using the standard causal mask or the block causal mask introduced in Section 3.6), the TWM returns a sequence of output embeddings:

$$(E(q_1^1), \ldots, E(q_1^L), E(a_1), \ldots E(q_T^1), \ldots, E(q_T^L), E(a_T)).$$

The TWM then output embeddings are then used to decode the following predictions:

**(1)** Following (Micheli et al., 2022), $E(a_t)$ passes through a reward head and predicts the logits of the reward $r_t$.

**(2)** $E(a_t)$ also passes through a termination head and predicts the logits of the termination state $\text{done}_t$.

**(3)** Without block teacher forcing, $E(q_t^i)$ passes through an observation head and predicts the logits of the next codebook index at the same timestep $E(q_t^{i+1})$, when $t \leq L - 1$. Similarly $E(a_t)$ passes through an observation head and predicts the logits of the first codebook index at the next timestep $E(q_{t+1}^1)$.

**(3')** With block teacher forcing, $E(q_t^i)$ passes through an observation head and predicts the logits of the same codebook index at the next timestep $E(q_{t+1}^i)$.

TWM is then trained by summing three losses:

**(1)** The first loss is the cross-entropy for the reward prediction. Note that Craftax-classic provides a (sparse) reward of 1 for the first time each achievement is "unlocked" in each episode. In addition, it gives a smaller (in magnitude) but denser reward, penalizing the agent by 0.1 for every point of damage taken, and rewarding it by 0.1 for every point recovered. However, we found that we got better results by ignoring the points damaged and recovered, and using a binary reward target, which we implemented by setting the target reward to 1 when the reward collected is higher than 0.5, and to 0 otherwise. This is similar to the recommendations in Farebrother et al. (2024), where the authors show that value-based RL methods work better when replacing MSE loss for value functions with cross-entropy on a quantized version of the return.

**(2)** The second loss is the cross-entropy for the termination predictions.

**(3** The third loss is the cross-entropy for the codebook predictions, where the predicted codes vary between 1 and the codebook size $K$.

**Architecture:**  We use the standard GPT2 architecture (Radford et al., 2019). We use a sequence length $T_{\text{WM}} = 20$ due to memory constraints. We implement key-value caching to generate rollouts fast. Table 6 details the different hyperparameters.

Table 6: Hyperparameters for the transformer world model

| Module | Hyperparameter | Value |
|---|---|---|
| Environment | Sequence length $T_{\text{WM}}$ | 20 |
| Architecture | Embedding dimension | 128 |
| | Number of layers | 3 |
| | Number of heads | 8 |
| | Mask | Causal or Block causal |
| | Inference with key-value caching | True |
| | Positional embedding | RoPE (Su et al., 2024) |
| Learning | Embedding dropout | 0.1 |
| | Attention dropout | 0.1 |
| | Residual dropout | 0.1 |
| | Optimizer | Adam (Kingma, 2014) |
| | Learning rate | 0.001 |
| | Max. gradient norm | 0.5 |

## A.3. Our Model-based RL agent

In this section, we detail how we combine the different modules above to build our SOTA MBRL agent, which is described in Algorithm 1 in the main text.

### A.3.1. COLLECTING ENVIRONMENT ROLLOUT OR TWM ROLLOUT

Algorithm 4 presents the rollout method, which we call in Steps 1 and 4 of Algorithm 1. It requires a transition function which can either be the environment or the TWM.

---

**Algorithm 4** Environment rollout or TWM rollout

---

**Input:** Initial observation $O_1$,
Previous $M$ observations $O_{\text{past}} = (O_{-M+1}, \ldots, O_0)$ if available else $O_{\text{past}} = \emptyset$,
AC model $\pi$ and parameters $\Phi$,
Rollout horizon $T$,
An environment transition $\mathcal{M}_{\text{env}}$ or a TWM $\mathcal{M}$ with parameters $\Theta$.

**Output:** A trajectory $\tau = (O_{1:T+1}, a_{1:T}, r_{1:T}, \text{done}_{1:T}, h_{0:T})$

**Initialize:** hidden state $h_0 = 0$ if $O_{\text{past}} = \emptyset$ else set $h_{-M} = 0$
**if** $O_{\text{past}} \neq \emptyset$ **then**
    *// Burn-in the hidden state*
    **for** $m = 1$ **to** $M$ **do**
        $z_{-M+m} = \text{ImpalaCNN}_\Phi(O_{-M+m})$
        $h_{-M+m} = \text{RNN}_\Phi([h_{-M-1+m}, z_{-M+m}])$
    **end for**
**end if**

**Initialize:** $\tau = (h_0)$

**for** $t = 1$ **to** $T$ **do**
    *// Run the actor network*
    $z_t = \text{ImpalaCNN}_\Phi(O_t)$
    $h_t = \text{RNN}_\Phi([h_{t-1}, z_t])$
    $a_t \sim \pi_\Phi([h_t, z_t])$

    *// Collect reward and next observation*
    **if** environment rollout **then**
        $O_{t+1}, r_t, \text{done}_t \sim \mathcal{M}_{\text{env}}(O_t, a_t)$
    **else if** TWM rollout **then**
        $Q_t = (q_t^1, \ldots, q_t^L) = \text{enc}(O_t)$
        $Q_{t+1} \sim p_\Theta(Q_{t+1}|Q_{1:t}, a_{1:t})$
        $O_{t+1} = \text{dec}(Q_{t+1})$
        $r_t \sim p_\Theta(r_t|Q_{1:t}, a_{1:t})$
        $\text{done}_t \sim p_\Theta(\text{done}_t|Q_{1:t}, a_{1:t})$
    **end if**
    $\tau + = (O_t, a_t, r_t, \text{done}_t, h_t)$
**end for**
$\tau + = (O_{T+1})$

---

Below we discuss various components of Algorithm 4.

**Parallelism.** Note that in Algorithm 1, we call Algorithm 4 in parallel starting from $N_{\text{env}}$ observations $O_1^{1:N_{\text{env}}}$ (for environment rollouts) or $\tilde{O}_1^{1:N_{\text{env}}}$ (for TWM rollouts).

**Burn-in.** The first time we collect data in the environment, we initialize the hidden state to zeros. The next time, we use burn-in to refresh the hidden state before rolling out the policy (Kapturowski et al., 2018). We do so by passing the $M$

observations prior to $O_1$ to the policy, which updates the hidden state of the policy using the latest parameters. (To use burn-in TWM rollout, we sample a trajectory of length $M + 1$ in Step 4 of Algorithm 1.) To enable burn-in, when collecting data, in Step 1 of Algorithm 1, we must also store the last $M$ environment observations $(O_{-M+1}, \ldots, O_0)$ prior to $O_1$.

**TWM sampling.**  As explained in the main text, sampling from the distribution $Q_{t+1} \sim p_\Theta(Q_{t+1}|Q_{1:t}, a_{1:t})$ is different when using (or not) block teacher forcing. For the former, the tokens of the next timestep $(q_{t+1}^1, \ldots, q_{t+1}^L)$ are sampled in parallel, while for the latter, they are sampled autoregressively.

**Maximum buffer size.**  To avoid running out of memory, we use a maximum buffer size and restrict the data buffer $\mathcal{D}$ in Algorithm 1 to contain at most the last 128k observations. When the buffer is at capacity, we remove the oldest observations before adding the new ones. We use flashbax (Toledo et al., 2023) to implement our replay buffer in JAX.

### A.3.2. WORLD MODEL UPDATE

In practice, we decompose the world model updates into two steps. First, we update the tokenizer $N_{\text{tok}}^{\text{iters}}$ times. Second, we update the TWM $N_{\text{TWM}}^{\text{iters}}$ times. For both updates, we use $N_{\text{WM}}^{\text{mb training}} = 3$ minibatches. That is, Step 3 of Algorithm 1 is implemented as in Algorithm 5.

---

**Algorithm 5** Step 3 of Algorithm 1

---

**for** it $= 1$ **to** $N_{\text{tok}}^{\text{iters}}$ **do**
  **for** $k = 1$ **to** $N_{\text{WM}}^{\text{mb training}}$ **do**
    $N^{\text{start}} = (k - 1)\left(N_{\text{env}}/N_{\text{WM}}^{\text{mb training}}\right) + 1$,  $N^{\text{end}} = k\left(N_{\text{env}}/N_{\text{WM}}^{\text{mb training}}\right) + 1$
    $\tau_{\text{replay}}^n = \text{sample-trajectory}(\mathcal{D}, T_{\text{WM}})$,              $n = 1 : N_{\text{env}}$
    $\Theta = \text{update-tokenizer}(\Theta, \tau_{\text{replay}}^{N^{\text{start}}:N^{\text{end}}})$ with Equation (5)
  **end for**
**end for**
**for** it $= 1$ **to** $N_{\text{TWM}}^{\text{iters}}$ **do**
  **for** $k = 1$ **to** $N_{\text{WM}}^{\text{mb training}}$ **do**
    $N^{\text{start}} = (k - 1)\left(N_{\text{env}}/N_{\text{WM}}^{\text{mb training}}\right) + 1$,  $N^{\text{end}} = k\left(N_{\text{env}}/N_{\text{WM}}^{\text{mb training}}\right) + 1$
    $\tau_{\text{replay}}^n = \text{sample-trajectory}(\mathcal{D}, T_{\text{WM}})$,              $n = 1 : N_{\text{env}}$
    $\Theta = \text{update-TWM}(\Theta, \tau_{\text{replay}}^{N^{\text{start}}:N^{\text{end}}})$ following Appendix A.2.2
  **end for**
**end for**

---

We always set $N_{\text{TWM}}^{\text{iters}} = 500$ to perform a large number of gradient updates. For M1-3, we set $N_{\text{tok}}^{\text{iters}} = 500$ as well, but for M5 we reduce it to $N_{\text{tok}}^{\text{iters}} = 25$ for the sake of speed—since NNT only adds new patches to the codebook.

### A.3.3. PPO POLICY UPDATE

Finally, the PPO-policy-update procedure called in Steps 1 and 4 of Algorithm 1 follows Algorithm 3.

When using PPO for MBRL, we found it critical to use different numbers of minibatches and different numbers of epochs on the trajectories collected on the environment and with TWM.

In particular, as the trajectories collected in imagination are longer, we reduce the number of parallel environments, and use $N_{\text{env}}^{\text{mb}} = 8$ and $N_{\text{WM}}^{\text{mb}} = 1$. This guarantees that the PPO updates are on batches of comparable sizes—$6 \times 96$ for real trajectories, and $48 \times 20$ for imaginary trajectories.

In addition, while the environment trajectories are limited, we can simply rollout our TWM to collect more imaginary trajectories. Consequently, we set $N_{\text{env}}^{\text{epoch}} = 4$, and $N_{\text{WM}}^{\text{epoch}} = 1$.

Finally, we do not use learning rate annealing for MBRL training.

## A.3.4. HYPERPARAMETERS

Table 7 summarizes the main parameters used in our MBRL training pipeline.

Table 7: MBRL main parameters

| Hyperparameter | Value |
| --- | --- |
| Number of environments $N_{\text{env}}$ | 48 |
| Rollout horizon in environment $T_{\text{env}}$ | 96 |
| Rollout horizon for TWM $T_{\text{WM}}$ | 20 |
| Burn-in horizon $M$ | 5 |
| Buffer size | $128,000$ |
| Number of tokenizer updates $N_{\text{tok}}^{\text{iters}}$ (with VQ-VAE) | 500 |
| Number of tokenizer updates $N_{\text{tok}}^{\text{iters}}$ (with NNT) | 25 |
| Number of TWM updates $N_{\text{TWM}}^{\text{iters}}$ | 500 |
| Number of minibatches for TWM training $N_{\text{WM}}^{\text{mb training}}$ | 3 |
| Background planning starting step $T_{\text{BP}}$ | 200k |
| Number of policy updates $N_{\text{AC}}^{\text{iters}}$ | 150 |
| Number of PPO minibatches in environment $N_{\text{env}}^{\text{mb}}$ | 8 |
| Number of PPO minibatches in imagination $N_{\text{WM}}^{\text{mb}}$ | 1 |
| Number of epochs in environment $N_{\text{env}}^{\text{epoch}}$ | 4 |
| Number of epochs in imagination $N_{\text{WM}}^{\text{epoch}}$ | 1 |
| Learning rate annealing | False |

## B. Comparing scores

Figure 9 completes the two main Figures 1 and 4 by reporting the scores the different agents. Specifically, Figure 9[left] compares our best MBRL and MFRL agents to the best previously published MBRL and MFRL agents. Figure 9[right] displays the scores for the different agents on our ladder of improvements.

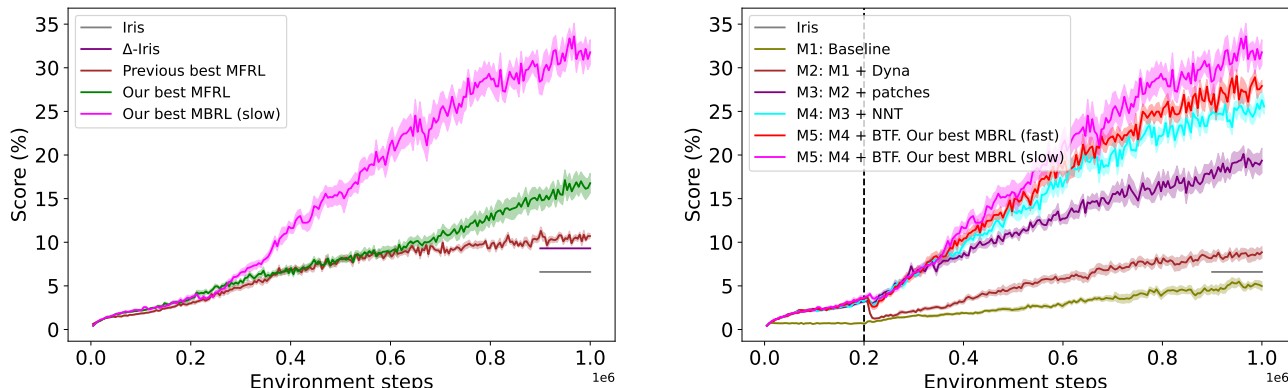

Figure 9: [Left] In addition to reaching higher rewards, our best MBRL and MFRL agents also achieve higher scores compared to the best previously published MBRL and MFRL results. [Right] MBRL agents' scores increase as they climb up the ladder of improvements.

## C. Annealing the number of policy updates

Figure 10 compares our best MFRL agent (with fast training) to an agent trained by annealing the number of policy updates in imaginary rollouts.

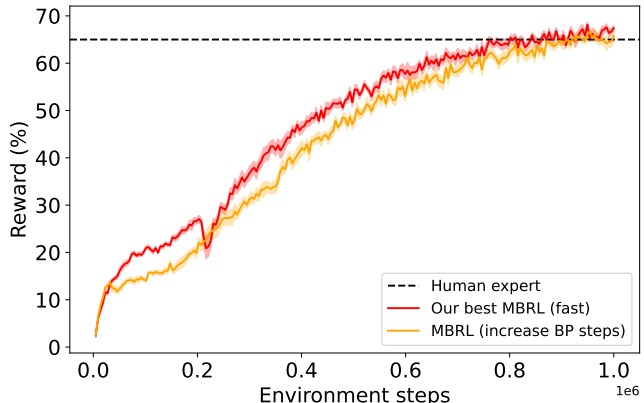

Figure 10: Progressively increasing the number of policy updates from $N_{\text{AC}}^{\text{iters}} = 0$ (when $T_{\text{total}} = 0$ env. steps) to $N_{\text{AC}}^{\text{iters}} = 300$ (when $T_{\text{total}} = 1M$) removes the drop in performance observed otherwise when we start training in imagination.

# D. Additional world model comparisons

This section complements Section 4.4 and presents two additional results to compare the different world models.

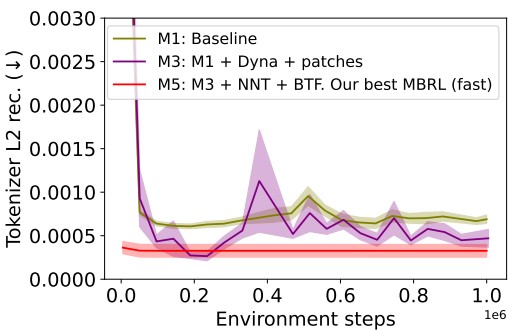 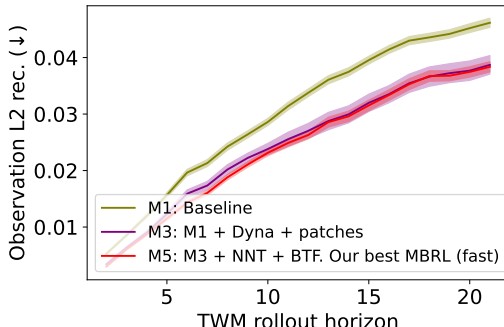

Figure 11: TWM performance.[Left] Tokenizer L2 reconstruction error, averaged over rollouts. Lower is better. By construction, our best MBRL agent, which uses NNT, constantly reaches the lowest error, as NNT directly adds observation patches to its codebook. [Right] TWM rollouts L2 observation reconstruction error, averaged over rollouts. Lower is better. M3 and M5, which both use patch factorization, achieve the lowest errors.

## D.1. Tokenizer reconstruction error

We first use the evaluation dataset $\mathcal{D}_{\text{eval}}$ (introduced in Section 4.4) to compare the tokenizer reconstruction error of our world models M1, M3, and M5—using the checkpoints at 1M steps. To do so, we independently encode and decode each observation $O_t^n \in \mathcal{D}_{\text{eval}}$, to obtain a tokenizer reconstruction $\hat{O}_t^{\text{tok}, n}$. Figure 11[left] compares the average L2 reconstruction errors over the evaluation dataset:

$$\frac{1}{(T+1)N_{\text{eval}}} \sum_{n=1}^{N_{\text{eval}}} \sum_{t=1}^{T_{\text{eval}}+1} \|\hat{O}_t^{\text{tok}, n} - O_t^n\|_2^2,$$

showing that all three models achieve low $L2$ reconstruction error. However our best model M5, which uses NNT, reaches a very low reconstruction error from the first iterations, since it directly adds image patches to its codebook rather than learning the codes online.

## D.2. Rollout reconstruction error

Second, given a sequence of observations in a TWM rollout $\hat{O}_{1:T_{\text{eval}}+1}^{\text{TWM}, n}$, and the corresponding sequence of observations in the environment $O_{1:T_{\text{eval}}+1}^n$ (which both have executed the same sequence of actions), Figure 11[right] compares the observation L2 reconstruction errors at each timestep $t$ (averaged over the evaluation dataset):

$$\mathcal{E}_t = \frac{1}{N_{\text{eval}}} \sum_{n=1}^{N_{\text{eval}}} \|\hat{O}_t^{\text{TWM}, n} - O_t^n\|_2^2, \quad \forall t.$$

As expected, the errors increase with $t$ as mistakes compound over the rollout. Our best method and M3, which both uses patch factorization, achieve the lowest reconstruction errors.

## D.3. Symbol extractor architecture

Herein, we discuss the symbol extractor architecture introduced in Section 4.4. $f_\mu$ consists of (a) a first convolution layer with kernel size $7 \times 7$, stride of 7, and channel size 128, which extracts a feature for each patch, (b) a ReLU activation, (c) a second convolution layer with kernel size $1 \times 1$, a stride of 1, and a channel size 128, (d) a second ReLU activation, (e) a final linear layer which transforms the 3D convolutional output into a 2D array of logits of size $145 * 17 = 1345$—where $R = 145$ is the number of ground truth symbols associated with each image of Craftax-classic and each symbol $S_t^r \in \{1, \ldots, 17\}$. The symbol extractor is trained with a cross-entropy loss between the predicted symbol logits and their ground truth values $S_t$, and achieves a $99.0\%$ validation accuracy.

### D.4. Rollout comparison

In Figure 12, we show an additional rollout that exhibits similar properties to those in Figure 6[right]. M1 and M3 make more simple mistakes in the map layout. All models generate predictions that can be inconsistent with the game dynamics. However the errors by M1 and M3 are more severe, as M5's mistake relates to the preconditions of the make sword action.

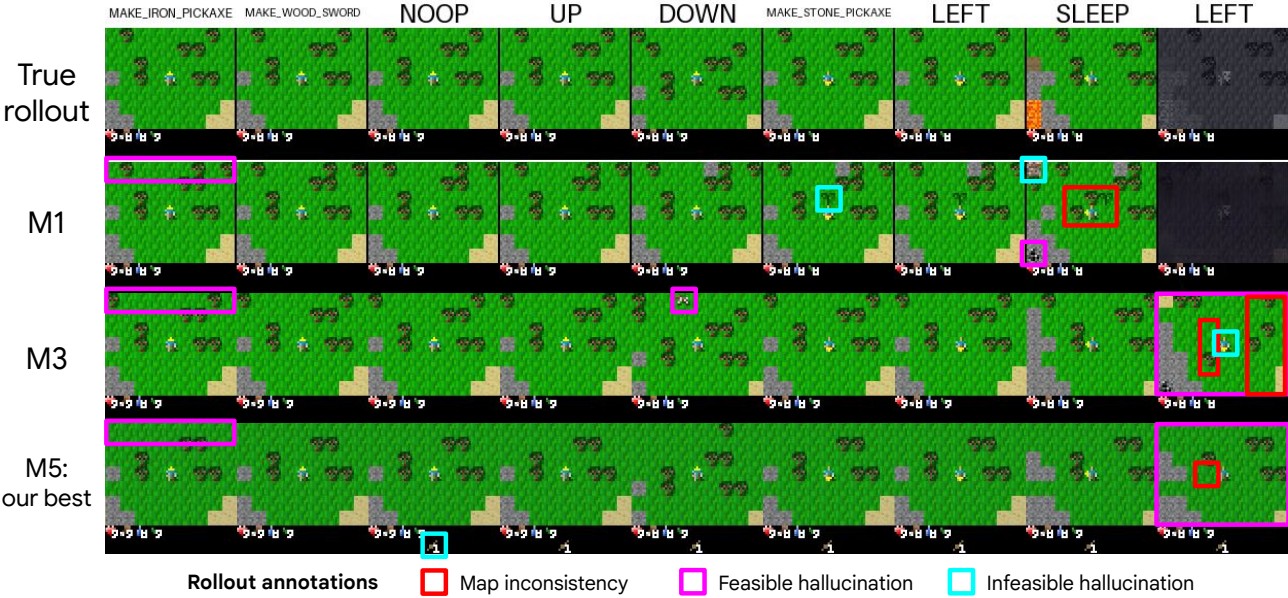

Figure 12: Additional rollout comparison for world models M1, M3 and M5. Best viewed zoomed in. **Map.** All models exhibit some map inconsistencies. M1 can make simple mistakes after the agent moves. Both M3 and M5 have map inconsistencies after the sleep actions, however the mistakes for M3 are far more severe. **Feasible hallucinations.** All models make feasible hallucinations when the agent exposes a new map region. The sleep action is stochastic, and only sometimes results in the agent sleeping after taking the action. As a result, M3 and M5 make reasonable generations in predicting that the agent does not sleep in the final frame. **Infeasible hallucinations.** M1 generates cells that do not respect the game dynamics, such as spawning a plant without taking the place plant action, and creating a block type that cannot exist in that location. M3 turns the agent to face downwards without the down action. M5 makes the wood sword despite the precondition of having wood inventory not being satisfied.

# E. Comparing Craftax-classic and Craftax (full)

This section complements Section 4.5 and discusses the main differences between Craftax-classic and Craftax.

The first and second block Table 8 compares both environments. Note that we only use the first five parameters in our experiments in Section 4.5.

The third and fourth blocks report the parameters used by our best MFRL and MBRL agents. In Craftax (full), for MFRL, we use $N_{\text{env}} = 64$ environments and a rollout length $T_{\text{env}} = 64$. Our SOTA MBRL agent uses $T_{\text{env}} = 96$, $N_{\text{env}} = 48$, and and $T_{\text{WM}} = 20$ as in Craftax-classic. We reduced the buffer size to 48k to fit in GPU.

All the others PPO parameters are the same as in Table 5.

Table 8: Environment Craftax-classic vs Craftax (full)

| Module | Hyperparameter | Classic | Full |
|---|---|---|---|
| Environment (used) | Image size | $63 \times 63$ | $130 \times 110$ |
| | Patch size | $7 \times 7$ | $10 \times 10$ |
| | Grid size | $9 \times 9$ | $13 \times 13$ |
| | Action space size | 17 | 43 |
| | Max reward (# achievements) | 22 | 226 |
| Environment (not used) | Symbolic (one-hot) input size | 1345 | 8268 |
| | Max cardinality of each symbol | 17 | 40 |
| | Number of levels | 1 | 10 |
| MFRL parameters | Number of environments $N_{\text{env}}$ | 48 | 64 |
| | Rollout horizon in environment $T_{\text{env}}$ | 96 | 64 |
| MBRL parameters | Number of environments $N_{\text{env}}$ | 48 | 48 |
| | Rollout horizon in environment $T_{\text{env}}$ | 96 | 96 |
| | Rollout horizon for TWM $T_{\text{WM}}$ | 20 | 20 |
| | Rollout horizon for TWM $T_{\text{WM}}$ | 20 | 20 |
| | Buffer size | $48,000$ | $128,000$ |

# F. Adapting Craftax-classic parameters to solve MinAtar

This section details the adaptations we made to our pipeline for solving the MinAtar environments, presented in Section 4.6. First, Table 9 outlines the modifications to our MFRL agent. Notably, we incorporate layer normalization (Ba et al., 2016) and Swish activation function (Ramachandran et al., 2017) within the ImpalaCNN architecture. Furthermore, we found it beneficial for the actor and critic networks to share weights up to their distinct final linear layers. We also adjust some PPO hyperparameters.

Table 9: MFRL changes for MinAtar

| Module | Parameter | Craftax | Minatar |
|---|---|---|---|
| Environment | Image size | $63 \times 63 \times 3$ | $10 \times 10 \times K$ |
| ImpalaCNN | Normalization | Batch normalization | Layer normalization |
| | Activation | ReLU | Swish |
| | Shared network | False | True |
| PPO | $\gamma$ | 0.925 | 0.95 |
| | $\lambda$ | 0.625 | 0.75 |
| | PPO target discount factor $\alpha$ | 0.95 | 0.925 |

These modifications result in a solid MFRL agent, whose performance is detailed in Section 4.6. We then develop our MBRL agent on top by implementing the changes outlined in Table 10. Specifically, we decompose each MinAtar image into 25 patches of size $2 \times 2 \times K$ each. In addition, we increase (a) the number of TWM updates to from 500 to 2k and (b) the number of policy updates in imagination from 150 to 2k. Critically, to address the high cost of bad actions in certain games (e.g. Breakout), we assign a weight of 10 to the cross-entropy losses of the reward and of the done states. This strongly penalizes inaccurate predictions of terminal states in imaginary rollouts. Additionally, we observe a potential issue during training in imagination where the agent could collapse to output the same action consistently. To mitigate this "action collapse" and promote exploration, we increase the entropy coefficient in the imagination phase from 0.01 to 0.05.

Table 10: MBRL changes for MinAtar

| Module | Parameter | Craftax | Minatar |
|---|---|---|---|
| Tokenizer | Patch size | $7 \times 7 \times 3$ | $2 \times 2 \times K$ |
| | Grid size | $9 \times 9$ | $5 \times 5$ |
| Training | Number of policy updates $N_{\text{AC}}^{\text{iters}}$ | 150 | $2,000$ |
| | Number of TWM updates $N_{\text{TWM}}^{\text{iters}}$ | 500 | $2,000$ |
| | Termination and reward weight | 1 | 10 |
| | PPO entropy coeff. in imagination | 0.01 | 0.05 |

Note that all the MinAtar games use the same hyperparameters.

