# OpenReview forum: "Improving Transformer World Models for Data-Efficient RL"
_ICML.cc/2025/Conference — ICML 2025 poster_

### Official Review · Reviewer_EkAK · 2025-03-12

**Overall Recommendation:** 4

**Summary:**

This paper presents a technically sound and empirically robust contribution to MBRL, with clear innovations in tokenization and transformer training. While the Craftax-centric evaluation limits immediate generalizability, the methodological advancements (NNT, BTF) are likely to inspire follow-up work. The paper is suitable for acceptance provided the authors address the above concerns, particularly by:

**Claims And Evidence:**

Yes

**Essential References Not Discussed:**

NA

**Experimental Designs Or Analyses:**

Yes

**Methods And Evaluation Criteria:**

Yes

**Other Comments Or Suggestions:**

Key implementation details (e.g., GRU architecture, codebook initialization for NNT) are briefly described but lack specificity. Public code or an appendix with full hyperparameters would aid reproducibility.

The human expert baseline (65.0% reward) is derived from 100 episodes by 5 players, but the paper does not clarify how this was standardized (e.g., episode length, interaction constraints). More details would strengthen the comparison.

**Other Strengths And Weaknesses:**

+ This paper presents a technically sound and empirically robust contribution to MBRL, with clear innovations in tokenization and transformer training. While the Craftax-centric evaluation limits immediate generalizability, the methodological advancements (NNT, BTF) are likely to inspire follow-up work. The paper is suitable for acceptance provided the authors address the above concerns, particularly by:

+The incremental "ladder of improvements" (Table 1) and detailed ablation analyses (Table 2, Figure 5) clearly demonstrate the contribution of each component (Dyna with warmup, NNT, BTF). The sensitivity analysis of patch size and warmup duration provides actionable insights.
+The nearest-neighbor tokenizer (NNT) and block teacher forcing (BTF) are novel and well-motivated. NNT’s static codebook addresses non-stationarity in VQ-VAE training, while BTF’s parallel token prediction mitigates autoregressive drift. These innovations are backed by quantitative metrics (symbol accuracy, reconstruction error) and qualitative rollout comparisons (Figure 6).
+The MBRL agent’s training time (759 minutes on 8 H100 GPUs) is competitive compared to prior work (e.g., IRIS: 18330 minutes). The MFRL baseline’s efficiency (15 minutes) further underscores the practicality of the approach.


-The evaluation is restricted to Craftax-classic and Craftax Full. While the environment’s complexity is a strength, the paper does not validate whether the proposed techniques (e.g., patch-based NNT) generalize to other domains (e.g., 3D environments, non-grid-based tasks). A discussion of applicability to broader settings (e.g., Minecraft, Atari, robotics) would strengthen the contribution, since the required codebook’s size for NNT would increase significantly in first-person-view settings, which may affect the stability of training TWM. While the empirical results are compelling, the paper lacks theoretical analysis of why BTF or NNT improve performance. For instance, how does BTF’s block causal attention reduce compounding errors compared to autoregressive sampling? This paper states that BTF improves performance by returning a more accurate TWM compared to AR methods. However, the observed gains may instead stem from more efficient representations learned through future state reasoning, rather than solely from improved accuracy. This distinction is not explicitly validated, as the ablation study does not include a direct comparison of dynamic losses. A deeper connection to existing theory (e.g., bias-variance trade-offs in world models) would add rigor.
-The NNT assumes aligned patches (e.g., 7×7 grids), which aligns with Craftax’s design but may not generalize to environments with less structured observations (e.g., raw pixel inputs in Atari). In such cases, NNT may allocate excessive computational resources to decoding background information, potentially overlooking small, reward-relevant objects. The paper briefly acknowledges this limitation but does not explore workarounds (e.g., adaptive patching).
- While the paper compares to IRIS and DreamerV3, it omits discussion of concurrent MBRL advances (e.g., Diamond's diffusion models, TD-MPC2’s continuous control). A broader literature review would better contextualize the work.

**Questions For Authors:**

No

**Relation To Broader Scientific Literature:**

They would potentially be useful for real-world applications (e.g., robotics, automated exploration).

**Theoretical Claims:**

NA

---

> ### Author Rebuttal · Authors · 2025-03-28
>
> Thank you for your positive evaluation of our work. We address your questions and comments below:
>
> **Weaknesses **
>
> (1) As detailed in our rebuttal to Reviewer fn71, we successfully trained an MBRL agent on the MinAtar benchmark, reusing our core MBRL components with little hyperparameter tuning. This agent significantly outperformed a tuned MFRL agent in these environments, highlighting the potential transferability of our approach to other grid-world settings.
>
> (2) We appreciate your suggestion for a theoretical comparison between the autoregressive and block teacher forcing (BTF) settings. We agree that this would be a valuable avenue for future investigation.
>
> (3) To quantify the impact of BTF in dynamics learning, we measured the average cross-entropy loss of the observation tokens over the last 500,000 training steps. Our TWM without BTF achieves an average CE of 0.478 (\pm 0.04), while our best TWM with BTF reaches 0.432 (\pm 0.004). This difference in learning dynamics suggests that BTF facilitates learning, likely by enabling the reuse of intermediate computations for next token predictions.
>
> (4) We agree that our proposed nearest neighbor tokenizer (NNT) is likely best-suited for grid-world environments. For environments with raw pixel inputs, such as Atari and Procgen, we believe that alternative tokenization methods like VQ-VAE and its variants will be necessary. We are actively exploring this direction. Our other two methods, Dyna with warmup and Block Teacher Forcing, should remain applicable in these settings.
>
> (5) Please note that we discuss TD-MPC2 and Diamond in our related work Section 2, including in our footnote 2.
>
>
> **Other comments or suggestions**
>
> (1) We would like to direct your attention to Appendix A, specifically Tables 4, 5, and 6, where we have tried to present all the hyperparameters used in our MBRL pipeline.
>
> (2) The human expert results utilized in this work have been extracted from the original Crafter paper. As detailed in Section 4.4 of [1], this dataset comprises 100 gameplay episodes recorded from five human experts who were given game instructions and several hours of practice prior to recording.
>
>
> **References**  [1] Hafner, D. Benchmarking the spectrum of agent capabilities.
> arXiv preprint arXiv:2109.06780, 2021.

---

### Official Review · Reviewer_6wf5 · 2025-03-12

**Overall Recommendation:** 3

**Summary:**

This paper proposes an approach for model-based reinforcement learning (MBRL) to improve sample efficiency and performance on the challenging Craftax-classic benchmark.
The method includes three improvements for both policy and transformer world model (TWM), which are “Dyna with warmup”, “nearest neighbor tokenizer” and “block teacher forcing”.

**Claims And Evidence:**

Yes, the claims are supported by clear and convincing evidence.

**Essential References Not Discussed:**

One of the key contributions is the parallel prediction of observation tokens, but a similar parallel prediction method is also proposed, namely Algorithm 1 & 2 from REM [1] published in ICML 2024.

[1] Lior Cohen, et al. "Improving Token-Based World Models with Parallel Observation Prediction." ICML (2024).

**Experimental Designs Or Analyses:**

I have checked the experimental designs in Section 4 and Appendix B, C & D.

**Methods And Evaluation Criteria:**

Yes, the proposed method makes sense.

**Other Comments Or Suggestions:**

NA

**Other Strengths And Weaknesses:**

* Strengths
1. This paper is well-written with clear motivation.
2. The three changes are sensible and effective, achieving significant performance gains in a Craftax-classic environment.
3. The experiments are solid, especially the ablation of the components.

* Weaknesses
1. The method contains too many hyperparameters, and it is difficult to determine which hyperparameters are important, making it difficult to transfer to other environments.

**Questions For Authors:**

1. According to Figure 1, using only PPO outperforms other baseline MBRL methods, most of which are based on REINFORCE. What would happen if you also build upon REINFORCE?
2. I am a bit confused about the NNT implementation, it looks like in the worst case NNT leads to unbounded memory overhead. How do you avoid this problem?

If you respond to these questions and address these concerns, I'll be willing to raise the score.

**Relation To Broader Scientific Literature:**

The key contribution of this paper is the three improvements for model-based reinforcement learning with a transformer world model:

1. “Dyna with warmup” follows the classic Dyna [1] setting.

2. The “nearest neighbor tokenizer” method is an improvement on the VQ-VAE in IRIS [2].

3. The “block teacher forcing” method modifies the attention mask of the GPT structure so that it can predict tokens in parallel, in a similar way as in REM [3].

[1] Richard S. Sutton, et al. “Dyna, an integrated architecture for learning, planning, and reacting.” ACM Sigart Bulletin (1991).

[2] Ajay Mandlekar, et al. “Transformers are sample-efficient world models." ICLR (2023).

[3] Lior Cohen, et al. "Improving Token-Based World Models with Parallel Observation Prediction." ICML (2024).

**Theoretical Claims:**

The paper does not present any formal theoretical proofs.

---

> ### Author Rebuttal · Authors · 2025-03-28
>
> **Essential References Not Discussed:**
>
> We thank you for pointing out the REM reference: we will include it in our revised paper. You are right that both block teacher forcing (BTF) and REM predict all the next frame tokens jointly. However, while REM uses a retentive network, BTF is applicable to a broader range of transformer architectures. BTF achieves joint prediction through a modification of the causal mask and the supervision signal, as illustrated in our Figure 3 and detailed in Appendix A.2.2.
>
> **Weaknesses:**
>
> We acknowledge that our model-based RL approach, like many others in this domain, involves a significant number of hyperparameters due to the interaction of its various components: the actor-critic policy, the tokenizer, and the world model.
> To provide clarity, we have dedicated separate appendices to detail each of these components: Appendix A.1 for the actor-critic policy, Appendix A.2.1 for the tokenizer, and Appendix A.2.2 for the world model. Table 6, Appendix A.3.4 summarizes the main MBRL parameters which glue these parameters together.
>
> Our supplementary experiments on the MinAtar environments, detailed in our rebuttal to Reviewer fn71, indicate that the majority of our proposed pipeline transfers to these other grid-world environments.
>
>
> **Questions:**
>
> (1) We opted to primarily utilize PPO due to its well-established advantages in terms of stability and performance. PPO's clipped surrogate objective function mitigates the issues of large policy updates that can destabilize learning, a common problem with vanilla REINFORCE. Furthermore, PPO generally outperforms REINFORCE in complex environments.
>
> Given these advantages, we believe PPO provides a robust foundation for achieving high levels of performance in Craftax-classic. Consequently, we anticipate a decrease in performance if we were to substitute REINFORCE.
>
> (2) You are right that in the worst case, our nearest neighbor tokenizer (NNT) can lead to memory overhead. However, we found that, on average, on Craftax-classic, only 2,304 codes (out of 4,096) are being used by NNT.

---

### Official Review · Reviewer_8HUm · 2025-03-14

**Overall Recommendation:** 4

**Summary:**

The authors propose a number of improvements to dreamer-style MBRL to achieve SOTA performance on the Craftax benchmark.

**Claims And Evidence:**

The authors mention three main contributions:
- They show that training on both real and imagined data is better than solely on imagined data
- They embed image patches using nearest neighbors, while prior work embeds the entire image using a VQ-VAE
- They propose to use a block-teacher forcing objective instead of a standard log likelihood objective

They provide ample evidence to back their major claims.

The authors also make smaller claims I have a bit of trouble with.

> How-
ever, the near-deterministic nature of Atari games allows
agents to memorize action sequences without demonstrat-
ing true generalization (Machado et al., 2018)

Virtually all modern work on Atari uses the `frameskip` variants where this is not true.

> For the RNN, we find it crucial to ensure
the hidden state is low-dimensional, so that the memory is
forced to focus on the relevant bits of the past that cannot
be extracted from the current image

Is there any evidence for this claim? Just because a smaller hidden size works better does not imply this specific causation.

**Essential References Not Discussed:**

I think they covered most important references.

**Experimental Designs Or Analyses:**

The authors meticulously ablate each and every component they introduce. I am quite happy with the depth of their experiments.

**Methods And Evaluation Criteria:**

Craftax is an interesting, well-known, and difficult benchmark.

**Other Comments Or Suggestions:**

> we fo-
cus on the Crafter domain (Hafner, 2021)

incorrect citation

**Other Strengths And Weaknesses:**

The authors start from a simple existing model, and slowly build up tricks to reach a new SOTA. I enjoyed reading the paper, and I like that the authors ablate every single proposed change. Reaching SOTA on Craftax is a notable acheivement, as the task is open-ended and difficutl.

One major complaint I have is that the image codebook does not seem like a scalable approach for more complex problems. Yes, it works for small pixel observation spaces, but I imagine it would fail in more realistic tasks.

**Questions For Authors:**

Not using imagined trajectories for some $k$ timesteps seems like a clunky solution to the problem. Did you consider reweighting imagined sampled by some variable that anneals from 0.0 to 1.0 as training goes on? The drop in Figure 4 suggests a "hard" warmup could hurt training.

**Relation To Broader Scientific Literature:**

The authors provide three improvements to dreamer-style MBRL. I believe other researchers focusing on such methods can integrate the authors' methods into their own models.

**Theoretical Claims:**

They do not make theoretical claims

---

> ### Author Rebuttal · Authors · 2025-03-28
>
> We are pleased that you enjoyed reading the paper and that you think that reaching SOTA on Craftax is a notable achievement, on a well-known and difficult benchmark.
>
> **Claims and Evidence:**
>
> (1) You are correct in that Atari with sticky actions (frameskip) makes it stochastic.  We will remove our claim of Atari being a deterministic environment.
>
> (2) We acknowledge that we do not demonstrate a causal relation between the low dimension of the hidden state, and the fact that the policy captures control relevant information. However, we varied the ratio of the RNN state dimension to the CNN encoder dimension, and found that a lower ratio yielded better performance, a result that contrasts with prior work. We will clarify this point in our revised paper.
>
> **Weaknesses:**
>
> We anticipate our proposed nearest neighbor tokenizer (NNT) to only work in grid-world environments. In environments with raw pixel inputs (Atari, Procgen) we believe that alternative tokenization methods, such as VQ-VAE and variants, are likely necessary. We are currently exploring this direction for future publication.
>
> **Questions:**
>
> Regarding annealing, we have conducted some follow-up experiments where we progressively increased the number of policy updates on imaginary rollouts ($N_{\text{AC}}^{\text{iters}}$ in Step 4 of Algorithm 1) from 0 to 300. This annealing technique achieves a reward of $65.7$% $(\pm1.11)$, while removing the drop in performance observed when we start training in imagination at $200$k steps. We will include these results in the revised paper.

---

### Official Review · Reviewer_fn71 · 2025-03-14

**Overall Recommendation:** 4

**Summary:**

The authors propose a new model-based RL method that achieves SOTA at Crafter. The superiority of there method stems from a variety of novel insights:
- adding a memory with low-dimensional hidden states and passing both the image embedding and the memory output to subsequent networks
- training on a mix of real and imaginary trajectories
- encoding image patches via nearest neighbor
- use of block teacher forcing to train the model

**Claims And Evidence:**

They perform an evaluation by incrementally adding there improvements on top of the baseline. There best model beats the baseline by more than 35% which is very significant.

**Essential References Not Discussed:**

N/A

**Experimental Designs Or Analyses:**

They use 10 seeds which is standard in RL.

**Methods And Evaluation Criteria:**

They evaluate there algorithm on Crafter which is a good benchmark.

**Other Comments Or Suggestions:**

N/A

**Other Strengths And Weaknesses:**

Strengths: The paper explain precisely the different design choices they made and they perform an extensive ablation study.
Weakness: Even if crafter is a good environment, I would like to see one or two more environments to assess the generality of the method. I am happy to raise my score once I see result for another environment.

**Questions For Authors:**

N/A

**Relation To Broader Scientific Literature:**

The authors build on the recent line of research on transformer world models and equip them with classical insights such as Dyna. Furthermore they propose a simpler alternative to VQ-VAE.

**Theoretical Claims:**

N/A

---

> ### Author Rebuttal · Authors · 2025-03-28
>
> We appreciate your favorable comments regarding our work.
>
> To further validate the robustness of our approach, we have conducted additional experiments on another grid-world environment MinAtar (https://github.com/kenjyoung/MinAtar), a set of four simplified Atari 2600 games. MinAtar contains symbolic binary observations of size 10x10xK, where K is the number of objects in each game, and binary rewards.
>
> We first tuned our model-free RL agent on these environments, keeping the same architecture as described in our paper, with minor adjustments to the PPO hyperparameters. Second, we developed our model-based RL agent, building upon our previously proposed techniques: Dyna warmup, nearest neighbor tokenizer, and block teacher forcing. We retained the majority of the MBRL hyperparameters from Craftax-classic, with the following key modifications: (a) we increased the number of WM training steps $N_{\text{TWM}}^{\text{iters}}$ to from 500 to 2k, (b) we increased the number of policy updates in imagination $N_{\text{AC}}^{\text{iters}}$  from 150 to 2k, (c)  we used patches of size 2x2xK (d) we added a weight of 10 to the cross-entropy losses of the reward and of the done states.
>
>
> The table below compares the performance of our agents, averaged over 10 seeds after 1 million environment steps.
>
> | Game | MFRL | MBRL  |
> |--------------|-----------------|-----------------|
> | Asterix | $10.28 \pm 1.38$ | $47.24 \pm 10.12$ |
> | Breakout | $76.36 \pm 1.82$ | $82.8 \pm 8.9$ |
> | Freeway | $61.33 \pm 3.67$ | $70.73 \pm 0.33$ |
> | SpaceInvader | $135.88 \pm 2.59$ | $180.7 \pm 3.34$ |
>
> These results appear quite competitive compared to existing approaches. Notably, our MBRL agent, after only 1M environment steps, seems to outperform the Artificial Dopamine method reported in Guan et al. [1] after 5M steps (as illustrated in their Figure 4).
>
> We are finalizing these experiments and plan to include them in the revised version of our paper.
>
>
> **Reference:**
> [1] Guan, Jonas, et al. "Temporal-Difference Learning Using Distributed Error Signals." Advances in Neural Information Processing Systems 37 (2024): 108710-108734.

---

### Decision · Program_Chairs · 2025-05-01

**Decision:**

Accept (poster)

**Comment:**

This paper presents a novel MBRL transformer-based algorithm that leverages nearest-neighbor image tokenization, hidden state memory, and real/imaged mixture of trajectories into training. The method provides solid motivations and insights into what led to success, which most notably was SOTA in Craftax environment, surpassing Dreamer V3.

There were concerns about hyperparameters and the limitations wrt the domains studied, but I feel like the contributions are clear enough to be a contribution other works can build on in other domains, and there is sufficient ablation studies to cover HP concerns.

I therefore recommend the paper accepted.